# Regulation of canonical Wnt signalling by the ciliopathy protein MKS1 and the E2 ubiquitin-conjugating enzyme UBE2E1

Katarzyna Szymanska[1], Karsten Boldt[2], Clare V Logan[1], Matthew Adams[1], Philip A Robinson[1†], Marius Ueffing[2], Elton Zeqiraj[3], Gabrielle Wheway[1,4,5]*, Colin A Johnson[1]*

[1]Leeds Institute of Medical Research, School of Medicine, University of Leeds, Leeds, United Kingdom; [2]Institute of Ophthalmic Research, Center for Ophthalmology, University of Tübingen, Tübingen, Germany; [3]Astbury Centre for Structural Molecular Biology, School of Molecular and Cellular Biology, Faculty of Biological Sciences, University of Leeds, Leeds, United Kingdom; [4]Faculty of Medicine, University of Southampton, Human Development and Health, UK, Southampton, United Kingdom; [5]University Hospital Southampton NHS Foundation Trust, Southampton, United Kingdom

**Abstract** Primary ciliary defects cause a group of developmental conditions known as ciliopathies. Here, we provide mechanistic insight into ciliary ubiquitin processing in cells and for mouse model lacking the ciliary protein Mks1. In vivo loss of Mks1 sensitises cells to proteasomal disruption, leading to abnormal accumulation of ubiquitinated proteins. We identified UBE2E1, an E2 ubiquitin-conjugating enzyme that polyubiquitinates β-catenin, and RNF34, an E3 ligase, as novel interactants of MKS1. UBE2E1 and MKS1 colocalised, and loss of UBE2E1 recapitulates the ciliary and Wnt signalling phenotypes observed during loss of MKS1. Levels of UBE2E1 and MKS1 are co-dependent and UBE2E1 mediates both regulatory and degradative ubiquitination of MKS1. We demonstrate that processing of phosphorylated β-catenin occurs at the ciliary base through the functional interaction between UBE2E1 and MKS1. These observations suggest that correct β-catenin levels are tightly regulated at the primary cilium by a ciliary-specific E2 (UBE2E1) and a regulatory substrate-adaptor (MKS1).

*For correspondence:
G.Wheway@soton.ac.uk (GW);
c.johnson@leeds.ac.uk (CAJ)

†Deceased

**Competing interest:** The authors declare that no competing interests exist.

## Editor's evaluation

It has long been recognized that ciliary dysfunction leads to increased canonical Wnt signaling but the mechanism has been elusive. Your work connecting β-catenin stability to Mks1 through Ube2e1 is an important advance in understanding this mechanism. I am certain that your work will stimulate more effort in this important area.

## Introduction

Primary cilia are microtubule-based organelles that sense and transduce extracellular signals on many mammalian cells. The cilium has essential roles throughout development during mechanosensation (*Praetorius and Spring, 2001*; *Nauli et al., 2003*), in transduction of multiple signalling pathways (*Huangfu et al., 2003*; *Simons et al., 2005*; *Schneider et al., 2005*) and in the establishment of left-right asymmetry (*Nonaka et al., 1998*). Primary cilia have a complex ultrastructure with compartmentalisation of molecular components that together form functional modules. Mutations in proteins that

are structural or functional components of the primary cilium cause a group of human inherited developmental conditions known as ciliopathies (*Adams et al., 2008*). Examples of ciliopathies include Meckel-Gruber syndrome (MKS) and Joubert syndrome (JBTS). Many proteins that are mutated in ciliopathies, including the MKS1 protein (*Szymanska and Johnson, 2012*; *Reiter et al., 2012*), localise to the transition zone (TZ), a compartment of the proximal region of the cilium. Mutations in the *MKS1* gene cause about 15% of MKS, a lethal neurodevelopmental condition that is the most severe ciliopathy (*Khaddour et al., 2007*).

The MKS1 protein contains a B9/C2 domain with homologies to the C2 (calcium/lipid-binding) domain of the synaptotagmin-like and phospholipase families (*Kyttälä et al., 2006*). MKS1 interacts with TMEM67, the transmembrane receptor encoded by the *TMEM67* gene (*Dawe et al., 2007*), and two other B9/C2-domain containing proteins, B9D1 and B9D2 (*Gupta et al., 2015*). B9D1, B9D2, and MKS1 are predicted to bind lipids in the ciliary membrane, and all three have been shown to localise at the ciliary TZ (*Bialas et al., 2009*) forming components of a functional module (known as the 'MKS-JBTS module'). This module contains other transmembrane proteins (TMEMs), namely the Tectonic proteins (TCTN1-3), TMEM17, TMEM67, TMEM231, and TMEM237, as well as other C2-domain proteins (jouberin, RPGRIP1L, and CC2D2A) (*Garcia-Gonzalo et al., 2011*; *Sang et al., 2011*; *Huang et al., 2011*). TZ proteins are thought to form a diffusion barrier at the base of the cilium that restricts entrance and exit of both membrane and soluble proteins (*Garcia-Gonzalo and Reiter, 2012*). The compartmentalisation of the cilium is essential for the regulated translocation of signalling intermediates, most notably during Sonic hedgehog (Shh) signalling (*Chih et al., 2011*), and mutations of TZ components invariably cause Shh signalling defects during development (*Weatherbee et al., 2009*). For example, mouse embryos from the *Mks1$^{Krc}$* knock-out mutant line have severe Shh signalling and left-right patterning defects during early embryonic development (*Weatherbee et al., 2009*). Previously, we have described the *Mks1* knock-out mouse line, for which mutant embryos have deregulated, increased canonical Wnt/β-catenin signalling and increased proliferation defects in the cerebellar vermis and kidney (*Wheway et al., 2013*).

Other studies have shown that the ciliary apparatus restricts the activity of canonical Wnt/β-catenin signalling (*Corbit et al., 2008*; *Lancaster et al., 2011*; *Simons et al., 2005*), although the mechanistic detail by which signal transduction is regulated remains unclear. One regulatory pathway involves the ciliary TZ protein jouberin (also known as AHI1), which shuttles β-catenin between the cytosol and nucleus in order to regulate Wnt signalling (*Lancaster et al., 2011*). However, ubiquitin-dependent proteasomal degradation by the ubiquitin-proteasome system (UPS) is the best-characterised mechanism for regulating canonical Wnt signalling (*Aberle et al., 1997*). In the absence of a Wnt signal, cytoplasmic β-catenin is phosphorylated in a complex of proteins (referred to as the destruction complex) that include axin, adenomatous polyposis coli (APC), and glycogen synthase kinase 3 (GSK-3)(*Ikeda et al., 1998*; *Munemitsu et al., 1995*; *Rubinfeld et al., 1996*). Subsequent ubiquitination of β-catenin leads to its degradation by the proteasome, meaning that in the absence of Wnt signalling the steady state levels of cytoplasmic β-catenin are low. Part of this regulation appears to be mediated by a functional association of the ciliary apparatus with the UPS (*Gerdes et al., 2007*), and UPS components have been shown to interact with ciliopathy proteins (e.g. USP9X and lebercilin) (*den Hollander et al., 2007*). RPGRIP1L (a ciliary TZ protein mutated in a range of ciliopathies including MKS and JBTS) has been reported to interact with the proteasome proteins, PSMD3 and PSMD5 (*Gerhardt et al., 2015*). Furthermore, discrete localisation of ubiquitin has been observed at the ciliary base suggesting that UPS processing can be constrained and regulated by the cilium (*Gerhardt et al., 2015*). However, the mechanistic basis to substantiate the association between the UPS and ciliary apparatus remains unclear and, in particular, it is unknown if the pathomechanism of Wnt signalling defects in ciliopathies depends on defective regulation of β-catenin localisation and processing by ciliary proteins.

Here, we describe the interaction and functional association of MKS1 with ciliary UPS components, specifically the E2 ubiquitin-conjugating enzyme UBE2E1 (also known as UbcH6) and the E3 ubiquitin ligating enzyme RNF34. In addition to ciliogenesis defects, loss of MKS1 causes deregulation of both proteasome activity and canonical Wnt/β-catenin signalling. These cellular phenotypes are also observed after loss of UBE2E1. MKS1 and UBE2E1 colocalise during conditions of cilia resorption, and levels of MKS1 and UBE2E1 are co-dependent. We show that in the absence of MKS1, levels of ubiquitinated proteins, including β-catenin, are increased. Furthermore, polyubiquitination of MKS1 is dependent on both UBE2E1 and RNF34, and lysine (Lys)63-linked polyubiquitination of MKS1 is

dependent on UBE2E1. This suggests that regulation of intracellular signalling, specifically canonical Wnt/β-catenin signalling, can be regulated and constrained at the primary cilium by a ciliary-specific E2 and MKS1, a substrate-adaptor.

## Results

### Mks1 mutation causes deregulation of proteasome activity

Loss of ciliary basal body proteins perturbs both UPS function and Wnt signalling (*Gerdes et al., 2007*), and we have previously reported de-regulated increases of canonical Wnt signalling in *Mks1⁻/⁻* mutant mice (*Wheway et al., 2013*). To investigate the mechanistic basis for regulation of canonical Wnt/β-catenin signalling and possible UPS processing of β-catenin by a ciliary protein, we first characterised these processes in cells and tissues lacking functional MKS1. We derived immortalised dermal fibroblasts from a human MKS patient, carrying compound heterozygous *MKS1* mutations [c.472C > T]+[IVS15-7_35del29] causing the predicted nonsense and splice-site null mutations [p.R158*]+[p. P470*fs*562] (*Khaddour et al., 2007*; *Figure 1—figure supplement 1a*) leading to loss of MKS1 protein (*Figure 1—figure supplement 1b-c*). *MKS1*-mutated fibroblasts had decreased cilia incidence and length (*Figure 1—figure supplement 1d*), and de-regulated canonical Wnt/β-catenin signalling (*Figure 1a*). *MKS1*-mutated fibroblasts had moderately increased levels of total β-catenin and the Wnt downstream target cyclin D1 (*Figure 1a*). SUPER-TOPFlash reporter assays confirmed that increased levels of β-catenin in *MKS1*-mutated fibroblasts caused de-regulated increases in canonical Wnt signalling in response to Wnt3a (a canonical Wnt ligand; *Figure 1b*). Treatment with the non-specific proteasome inhibitor MG-132 also increased levels of phosphorylated β-catenin (*Figure 1a*). Since β-catenin is phosphorylated to mark it for processing by the 26 S proteasome, we also tested if proteasome enzymatic activity was affected in *MKS1*-mutated fibroblasts. We observed increased proteasome activity, which was inhibited by treatment with lactacystin that targets the 20 S catalytic core of the proteasome, as well as moderate increased levels of the proteasome subunit α7 (*Figure 1c*). This was accompanied by increased levels of mono- and poly-ubiquitinated proteins in the *MKS1*-mutated fibroblasts following protease inhibition (*Figure 1—figure supplement 1e*).

To substantiate an in vivo association between de-regulated canonical Wnt signalling and proteasome activity in the ciliopathy disease state, we crossed the *Mks1* knock-out mouse line (*Wheway et al., 2013*) with the *Ub^{G76V}-GFP* transgenic reporter line. *Ub^{G76V}-GFP* constitutively degrades GFP-ubiquitinated proteins, leading to an absence of GFP signal if proteasome processing is unimpaired (*Lindsten et al., 2003*). Confirming our observations with human *MKS1*-mutated fibroblasts, *Mks1⁻/⁻* x *Ub^{G76V}-GFP* mouse embryonic fibroblasts (MEFs) also had de-regulated proteasome enzymatic activity (*Figure 1d*) compared to *Mks1⁺/⁺* x *Ub^{G76V}-GFP* wild-type littermate MEFs. Furthermore, after intraperitoneal injection of MG-262 into pregnant dams at E11.5, *Mks1⁻/⁻* x *Ub^{G76V}-GFP* mutant embryos at embryonic day E12.5 had increased levels of GFP, detected by both epifluorescence confocal microscopy and western blotting, in the neocortex (*Figure 1e*) and other tissues (*Figure 1—figure supplement 2*) compared to wild-type littermate controls. This suggests that in mutant mice abnormally high levels of polyubiquitinated proteins stimulate increased proteasome function (*Figure 1d*) that facilitates protein degradation and maintenance of correct levels of polyubiquitinated proteins in the cell. Upon proteasome inhibition, GFP-polyubiquitinated proteins accumulated in mutant mice tissues, indicating that there is a defect of abnormal increased protein polyubiquitination in mice lacking *Mks1*. Furthermore, these defects accompanied increased levels of active β-catenin in the neuroepithelium of *Mks1⁻/⁻* x *Ub^{G76V}-GFP* mutant ventricular zone (*Figure 1—figure supplement 2*).

### MKS1 interacts with the E2 ubiquitin-conjugation enzyme UBE2E1, with colocalisation during cilia resorption

To understand why mutation or loss of MKS1 causes de-regulated increases of both proteasome activity and canonical Wnt/β-catenin signalling, we sought to identify MKS1-interacting proteins. We performed a yeast two-hybrid screen using amino acids 144–470 of MKS1 that contain the B9/C2 domain as bait (*Figure 2a*) and identified the E2 ubiquitin conjugation enzyme UBE2E1 (also known as UbcH6) (*Hong et al., 2008*) as an interactant of MKS1 (*Figure 2b*). We confirmed this interaction by a 'one-to-one' yeast two-hybrid assay (*Figure 2c*). Additionally, we identified the E3 ubiquitin ligase RNF34 and confirmed its interaction and colocalisation with MKS1 (*Figure 2—figure supplement*

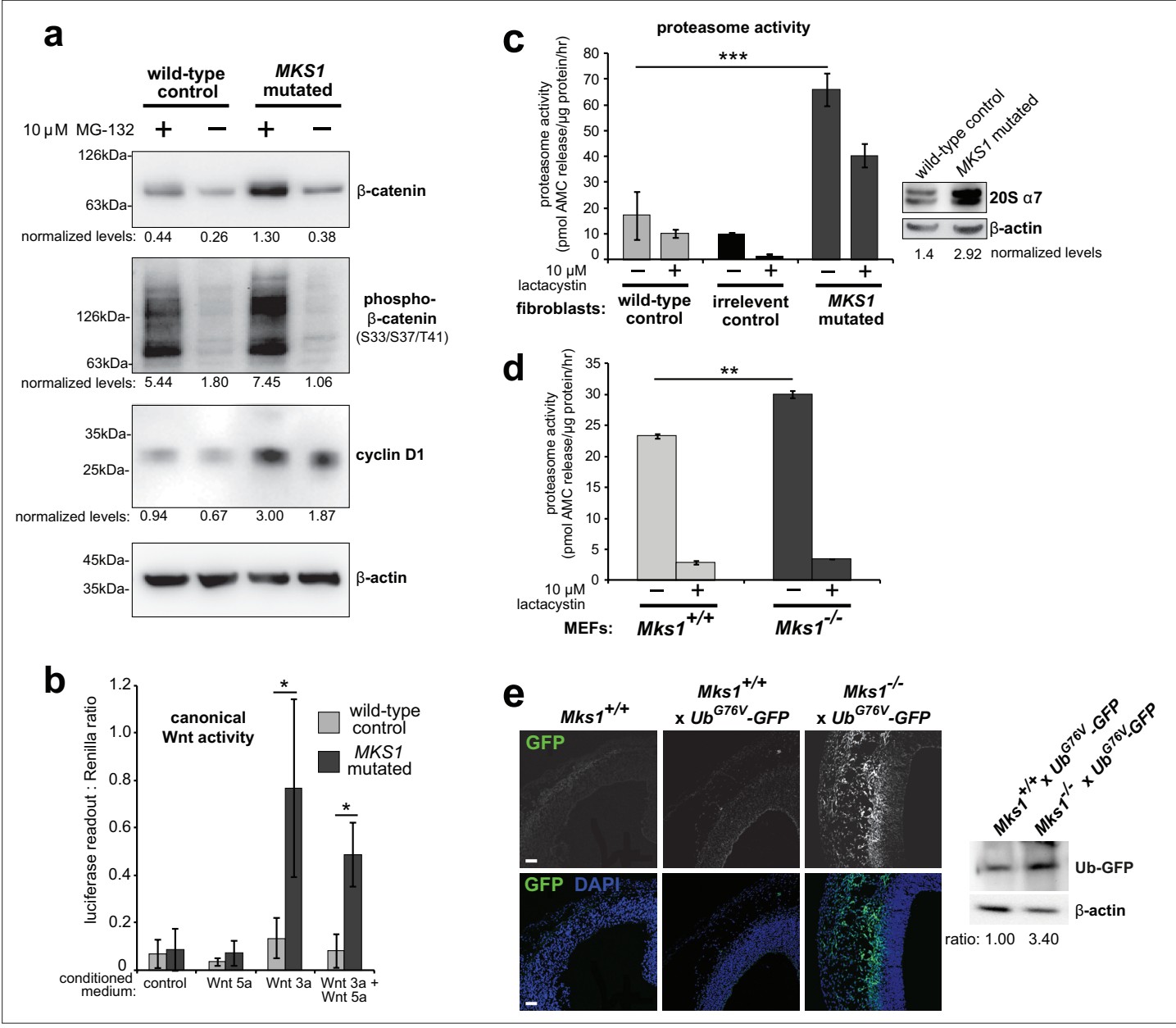

**Figure 1.** Deregulation of canonical Wnt signalling and proteasome activity following loss or mutation of MKS1. (**a**) Immunoblots for total soluble β-catenin, phospho-β-catenin, cyclin D1 and β-actin (loading control) in either wild-type normal or *MKS1*-mutated immortalised human fibroblasts from an MKS patient (MKS-562) following treatment with MG-132 proteasome inhibitor (+) or vehicle control (-). (**b**) SUPER-TOPFlash assays of canonical Wnt signalling activity in human *MKS1*-mutated fibroblasts compared to wild-type control fibroblasts following treatment with control conditioned medium, Wnt5a, Wnt3a, or a mixture of Wnt3a and Wnt5a media, as indicated. Statistical significance of pairwise comparisons is shown (* indicates p < 0.05, paired two-tailed Student t-test). Error bars indicate s.e.m. with results shown for four independent biological replicates. (**c**) Proteasome activity assays for wild-type or *MKS1*-mutated human fibroblasts or an irrelevant control (*ASPM*-mutant fibroblasts), following treatment with c-lactacystin-β-lactone (+) or vehicle control (-). Statistical significance of pairwise comparison as for (**b**); *** indicates p < 0.001 for three independent biological replicates. Immunoblots show levels of the 20 S proteasome α7 subunit compared to β-actin loading control. (**d**) Protease activity assays of crude proteasome preparations from *Mks1⁺/⁺* or *Mks1⁻/⁻* mouse embryonic fibroblasts (MEFs), expressed as pmol AMC released per μg proteasome per hr. Treatment with lactacystin is the assay control. Statistical analysis as for (**b**); ** indicates p < 0.01 for three independent biological replicates. (**e**) Accumulation of GFP-tagged ubiquitin (GFP; green) in *Mks1⁻/⁻* x *Ub^{G76V}-GFP* E12.5 embryonic cerebral neocortex treated with MG-262 proteasome inhibitor. Immunoblot for GFP in *Mks1⁻/⁻* x *Ub^{G76V}-GFP* and wild-type littermate E12.5 embryo protein lysates, with immunoblotting for β-actin as a loading control, showing accumulation of GFP-tagged ubiquitin (Ub-GFP) in *Mks1⁻/⁻*.

The online version of this article includes the following source data and figure supplement(s) for figure 1:

**Source data 1.** Characterisation of MKS1-mutated human patient fibroblasts: full western blots.

*Figure 1 continued on next page*

*Figure 1 continued*

**Figure supplement 1.** Characterisation of *MKS1*-mutated human patient fibroblasts.

**Figure supplement 1—source data 1.** Characterisation of MKS1-mutated human patient fibroblasts: full western blots & gels.

**Figure supplement 2.** In vivo loss of MKS1 causes deregulated ubiquitin-proteasome processing.

*1*). In support of a possible role of MKS1 in regulating ubiquitinated signalling proteins, UBE2E1 has been described to function as an E2 with the E3 JADE-1 during the ubiquitination of β-catenin (*Chitalia et al., 2008*). We therefore further substantiated the interaction of UBE2E1 with MKS1. We purified GST-tagged UBE2E1 and confirmed the interaction between MKS1 and UBE2E1 by a GST pull-down assay (*Figure 2d*). The interaction between endogenous MKS1 and UBE2E1 was confirmed by co-immunoprecipitations (co-IPs) using anti-MKS1 (*Figure 2e*). This was further corroborated when an interaction between endogenous UBE2E1 and exogenously expressed cmyc-tagged MKS1 was detected by co-IP with an anti-UBE2E1 antibody (*Figure 2f*).

## The UBE2E1-MKS1 interaction is required for cilia resorption

Having confirmed UBE2E1-MKS1 interaction in vitro and in cells, we next assessed if this was important for cilia function. UBE2E1 and MKS1 co-localised at the basal body in a subset of confluent, ciliated hTERT-immortalised retinal pigment epithelium RPE1 and ARPE19 cells during $G_0$ of the cell cycle following serum starvation for 48 hr (*Li et al., 2011 Figure 3a–d*). Serum starvation, followed by

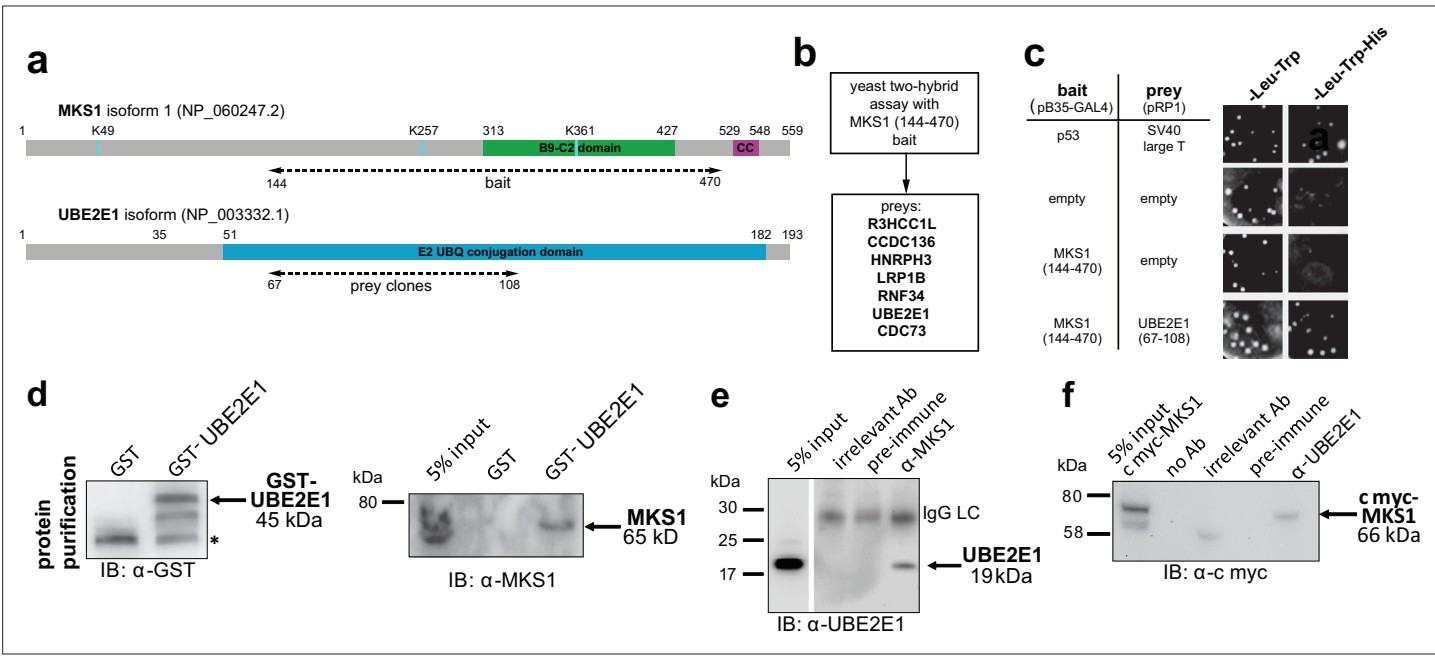

**Figure 2.** The E2 ubiquitin conjugation enzyme UBE2E1 interacts with MKS1. (**a**) Domain structure of MKS1 and UBE2E1 proteins for the indicated isoform showing the locations of the B9/C2 domain, putative ubiquitinated lysines in blue (predicted by UbPred), a predicted coiled-coil (CC) motif, and the E2 ubiquitin (UBQ) conjugation domain in UBE2E1. Numbering indicates the amino acid residue. Dashed lines indicate the region used as 'bait' in MKS1 for the yeast two-hybrid assay and the 'prey' clones in the UBE2E1 interactant. (**b**) List of preys identified in the MKS1 Y2H screen (**c**) Left panel: yeast 'one-to-one' assays for the indicated bait, prey and control constructs. Right panel: only colonies for the positive control (p53+ SV40 large T) and MKS1 bait+ UBE2E1 prey grew on triple dropout (-Leu -Trp -His) medium. (**d**) GST-UBE2E1 purified from bacterial extracts (left panel) pulled down endogenous MKS1 from ARPE19 whole cell extract. (**e**) Co-immunoprecipitation (co-IP) of endogenous UBE2E1 by rabbit polyclonal anti-MKS1, but not pre-immune serum or an irrelevant antibody (Ab; anti cmyc); IgG light chain (LC) is indicated. (**f**) Co-IP of exogenously expressed cmyc-MKS1 by anti-UBE2E1 but not pre-immune serum or an irrelevant antibody.

The online version of this article includes the following source data and figure supplement(s) for figure 2:

**Source data 1.** The E2 ubiquitin conjugation enzyme UBE2E1 interacts with MKS1: full western blots.

**Figure supplement 1.** The E3 ubiquitin ligase RNF34 interacts with MKS1 and co-localises at the basal body.

**Figure supplement 1—source data 1.** The E3 ubiquitin ligase RNF34 interacts with MKS1 and co-localizses at the basal body: full western blot.

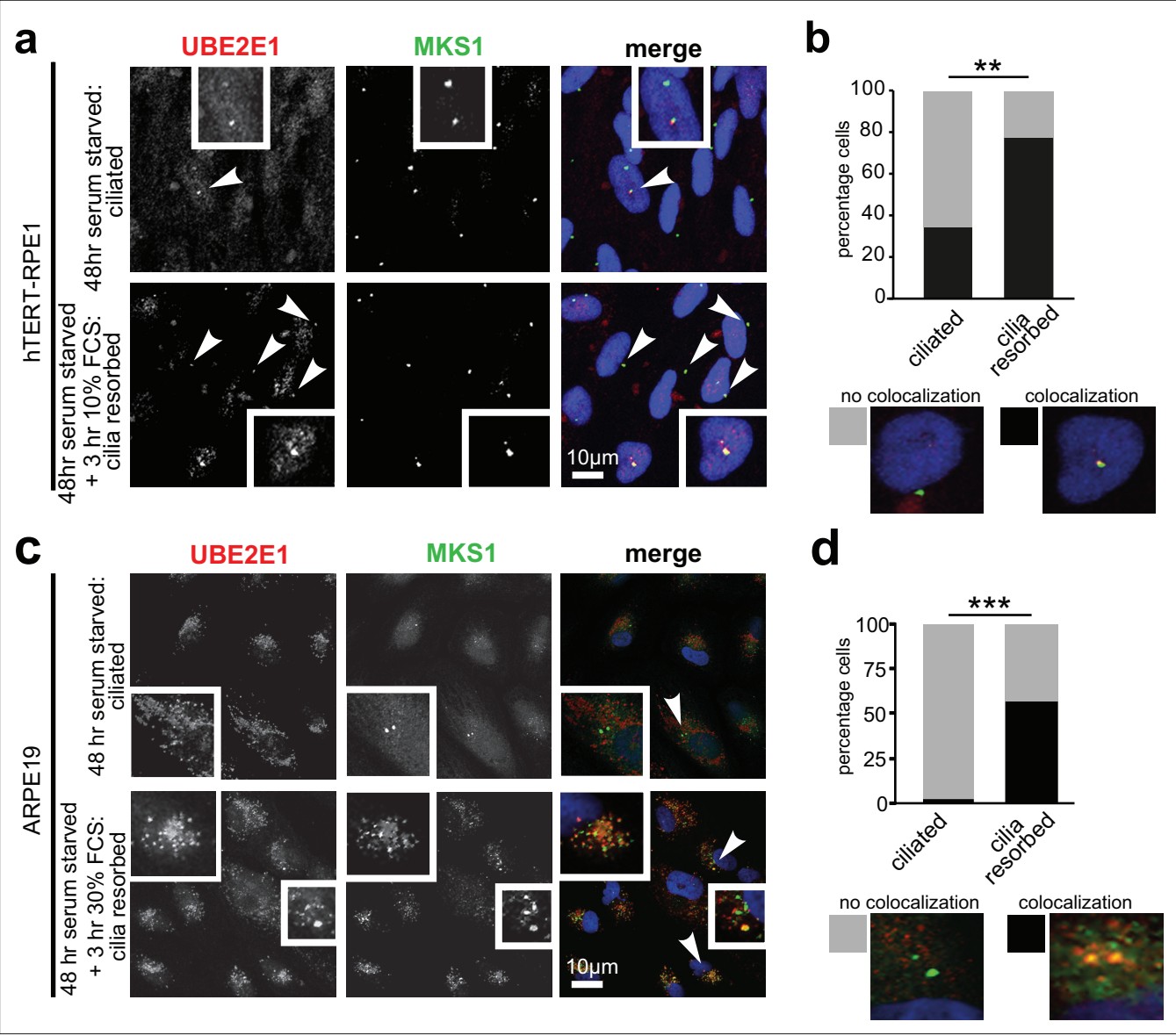

**Figure 3.** Co-localisation of MKS1 and UBE2E1 under conditions of ciliary resorption. (**a**) MKS1 (green) and UBE2E1 (red) partially colocalise at the basal body/centrosome in human wild-type hTERT-RPE1 cells, particularly when induced to resorb cilia by treatment with 10% FCS after 72 hr of serum starvation. White arrowheads indicate cells magnified in insets. Scale bar = 10 μm. (**b**) Bar graph indicates the percentage of cells in which MKS1 and UBE2E1 co-localise at the basal body (black), and the percentage without co-localisation (grey) for three independent biological replicates, with examples shown of representative cells. (**c**) Figure details as for (**a**) showing partial co-localisation of MKS1 and UBE2E1 in human ARPE19 cells. (**d**) Bar graph details as for (**b**). Data in (**b**) and (**d**) were analysed by two-way ANOVA followed by Tukey's multiple comparison test (statistical significance of comparisons indicated by ** p < 0.01, *** p < 0.001).

re-addition of serum for 3 hr, caused rapid cilia resorption (*Li et al., 2011*) with further significant colocalisation of UBE2E1 and MKS1 at the basal body (*Figure 3b and d*). This suggests that the interaction between MKS1 and UBE2E1 is particularly important during the process of cilia resorption.

## UBE2E1 mutation or loss causes ciliogenesis defects, and de-regulated increases in both proteasome activity and Wnt/β-catenin signalling

Since correct UPS function appears to be required for ciliogenesis (*Gerhardt et al., 2015*), we next asked if loss or mutation of UBE2E1 had an effect on ciliogenesis. UBE2E1 is an enzyme that transfers ubiquitin to a substrate, with or without the presence of an E3, in a reaction that is dependent on an active enzymatic domain. To assess if enzymatic activity of UBE2E1 is necessary for correct

ciliogenesis, we mutated the active site cysteine residue 131 to serine (*Nuber et al., 1996*) to make a dominant negative (DN) catalytically inactive form of UBE2E1. Over-expression of the Cys131Ser form of UBE2E1 caused significant loss and shortening of cilia in mouse inner medullary collecting duct (mIMCD3) cells (*Figure 4a–b*), suggesting that catalytically active UBE2E1 is required for normal ciliogenesis. Over-expression of wild-type (WT) UBE2E1 had a moderate dominant negative effect on cilia length only (*Figure 4a–b*).

To model the effect of UBE2E1 loss on ciliogenesis, we first used pooled and individual siRNA duplexes targeting *Ube2e1* in mIMCD3 cells. This affected ciliogenesis in mIMCD3 cells by reducing cilia incidence and length, but achieved only moderate knockdown of UBE2E1 protein levels (*Figure 4—figure supplement 1a*). To ensure more robust, long-term knockdown of UBE2E1, we derived stably-transfected mIMCD3 cell-lines with three different *Ube2e1* shRNA constructs. Each *Ube2e1* shRNA construct reduced UBE2E1 protein levels (compared to cells expressing scrambled shRNA), and significantly reduced both numbers of ciliated cells and mean cilium length (*Figure 4c–d*). To understand the effect loss of Ube2e1 had on Mks1, we pulled down Mks1 at different ciliogenesis stages in control cells and cells with stable knockdown of *Ube2e1* followed by identification of inter-acting proteins by LC-MS/MS mass spectrometry analysis (*Figure 4e,f*, *Figure 4—source data 1*). We observed significant decreases in peptide counts for sh*Ube2e1* knockdown cells across different conditions of ciliogenesis (*Figure 4—source data 1*; $\chi^2$ test p < 0.05), particularly under conditions of ciliary resorption (*Figure 4e*). Analysis for enrichment of GO terms identified specific biological processes that included 'cell-cell adhesion', 'ubiquitin-dependent protein catabolism', 'actin filament capping', and 'beta-catenin destruction complex' (*Figure 4f*). Interestingly, the 'actin filament capping' term includes known interactants of ciliopathy proteins such as filamin A (Flna) (*Adams et al., 2012*; *Figure 4—figure supplement 1b*).

Since UBE2E1 and MKS1 both interact and co-localise, we next determined if UBE2E1 loss reiter-ates the cellular phenotypes caused by MKS1 loss or mutation. Indeed, we observed increased prote-asome enzymatic activity compared to scrambled shRNA (shScr) negative control cells (*Figure 4g*). Furthermore, in agreement with *MKS1*-mutated fibroblasts and *Mks1*[-/-] MEFs, sh*Ube2e1* knock-down cells had de-regulated canonical Wnt/β-catenin signalling in response to Wnt3a (*Figure 4h*). This was accompanied by increased levels of mono- and poly-ubiquitinated proteins in sh*Ube2e1* knock-down cells following proteasome inhibition (*Figure 4—figure supplement 1c*), also consistent the effect observed in *MKS1*-mutated fibroblasts. These data highlight a possible important role of UBE2E1 in mediating correct protein ubiquitination, proteasome function and Wnt signalling in the context of cilia. The striking similarities in ciliary phenotypes suggest a close functional association between MKS1 and UBE2E1, and led us to hypothesise that they are placed in the same regulatory pathway.

## Mutual inhibition of MKS1 and UBE2E1 protein levels

To further investigate the possible functional association between MKS1 and UBE2E1, we tested if de-regulated Wnt signalling could be rescued by over-expression experiments. In a control experi-ment, expression of cmyc-tagged MKS1 partially rescued normal canonical Wnt signalling responses to Wnt3a in *MKS1*-mutated fibroblasts (*Figure 5a*). However, expression of FLAG-tagged UBE2E1 led to almost complete rescue of normal Wnt signalling responses (*Figure 5a*). Conversely, expression of MKS1 in sh*Ube2e1* knock-down cells also rescued canonical Wnt signalling (*Figure 5b*), suggesting co-dependency between MKS1 and UBE2E1. We confirmed this following transient siRNA knockdown of MKS1 that caused significantly increased levels of UBE2E1 in cells (*Figure 5c*). In the reciprocal experiment, MKS1 protein levels were significantly increased in sh*Ube2e1* knock-down cells, partic-ularly under conditions of ciliary resorption (*Figure 5c*). To further support co-dependency, we over-expressed both MKS1 and UBE2E1. At higher levels of UBE2E1, we observed a moderate decrease in MKS1 levels (*Figure 5d*). Conversely, expression of high levels of MKS1 caused a decrease in UBE2E1 protein levels (*Figure 5d*). These results show a striking co-dependency in protein levels between MKS1 and UBE2E1, suggesting inhibitory roles for each of these proteins on the protein level of the other.

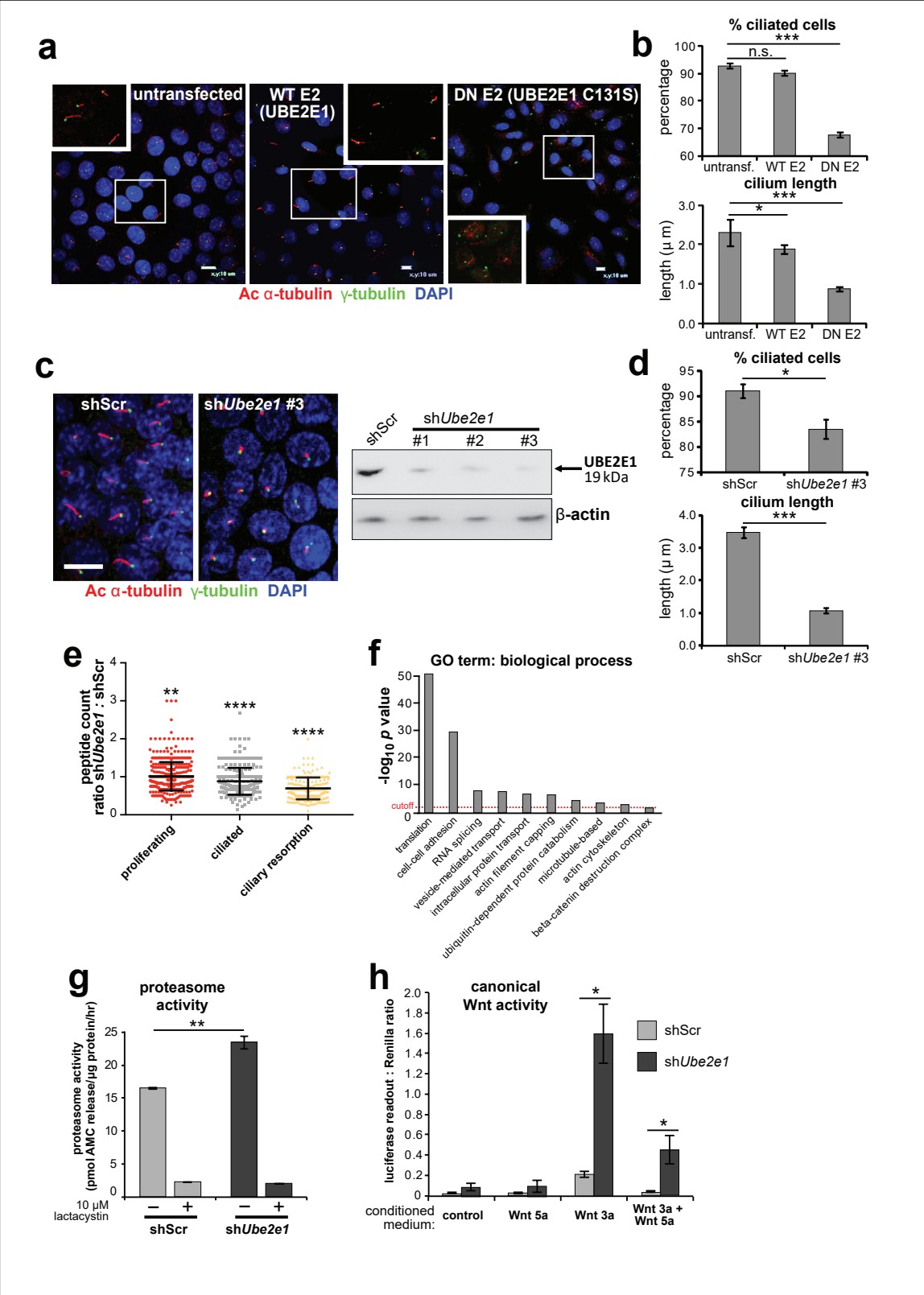

**Figure 4.** UBE2E1 is required for regulation of ciliogenesis, proteasome activity, and canonical Wnt signalling. (**a**) Primary cilia in mIMCD3 cells following transfection with either wild-type (WT) UBE2E1 (**E2**) or dominant negative (DN) UBE2E1 carrying the active site mutation C131S, compared to mock-transfected negative control. Scale bars = 10 μm. (**b**) For experiments shown in (**a**), statistical significance of pairwise comparisons with control (untransf.) for three independent biological replicates are shown (n.s. not significant, * p < 0.05, ** p < 0.01, *** p < 0.001; unpaired two-tailed Student t-test;

*Figure 4 continued on next page*

Figure 4 continued

error bars indicate s.e.m.). (**c**) shRNA-mediated knockdown of *Ube2e1* in stably-transfected mIMCD3 cell-line #3 causes decreased ciliary incidence and length. Scale bar = 10 µm. Immunoblot shows loss of UBE2E1 protein expression compared to β-actin loading control following shRNA knockdown. (**d**) Bar graphs quantifying decreased ciliary incidence and length with statistical analysis as for (**b**). (**e**) Scatter plot of relative differences in the proteins pulled-down by anti-MKS1 immunoprecipitations, under different conditions of ciliogenesis (proliferating cells, ciliated cells, cells undergoing ciliary resorption), expressed as the ratios of peptide counts for shScr: sh*Ube2e1* knockdowns. Statistical significance of pairwise comparisons for each set of ratios was calculated as for (**b**) (paired two-tailed Student t-tests). Error bars indicate s.d. Full data-sets are available in *Figure 4—source data 1*. (**f**) Bar graph of $-\log_{10} p$ values for significantly enriched GO terms (biological processes) for proteins included in (**e**), with cut-off for $p < 0.05$ indicated by the red dashed line. Enrichment for GO terms was analyzed by using DAVID (https://david.ncifcrf.gov/). (**g**) Protease activity assays of crude proteasome preparations from shScr and sh*Ube2e1* mIMCD3 knockdown cells, showing increased proteasomal activity in sh*Ube2e1* as assayed by pmol AMC released per µg proteasome per hour. Treatment with lactacystin is the assay control. Statistical significance of pairwise comparisons as for (**b**). (**h**) SUPER-TOPFlash assays of canonical Wnt signalling activity in sh*Ube2e1* cells compared to shScr following treatment with control conditioned medium, Wnt5a, Wnt3a, or a mixture of Wnt3a and Wnt5a media, as indicated. Statistical significance of pairwise comparisons of at least four independently replicated experiments as for (**b**).

The online version of this article includes the following source data and figure supplement(s) for figure 4:

**Source data 1.** Mass spectrometry results for MKS1 pull-downs from mIMCD3 cells across different conditions of ciliogenesis.

**Source data 2.** UBE2E1 is required for regulation of ciliogenesis, proteasome activity, and canonical Wnt signalling: full western blot.

**Figure supplement 1.** Validation of *Ube2e1* siRNA knockdown in mIMCD3 cells and effect of shRNA *Ube2e1* knockdowns on MKS1 interacting proteins under different conditions of ciliogenesis.

## MKS1 is polyubiquitinated and its polyubiquitination depends on UBE2E1

UBE2E1 is an E2 ubiquitin conjugating enzyme, and we next tested the obvious hypothesis that it participates in polyubiquitination and targeting MKS1 for degradation. We therefore investigated if MKS1 is indeed tagged with ubiquitin chains and if absence of UBE2E1 affects ubiquitination of MKS1. We determined MKS1 levels and its polyubiquitination status in different ciliogenesis conditions, namely: proliferating cells (grown in normal medium supplemented with serum); ciliated cells (quiescent cells grown in serum-deprived medium); and cells undergoing ciliary resorption (grown in serum-deprived medium, followed by serum re-addition for 3 hr). sh*Ube2e1* knock-down cells consistently had significantly increased levels of MKS1 as well as polyubiquitinated MKS1 (*Figure 6a*; $p < 0.05$ two-way ANOVA between shScr and sh*Ube2e1*). The highest levels of MKS1 were observed in cells undergoing cilia resorption, when MKS1 and UBE2E1 co-localisation is the strongest. Despite modest statistical significance, this finding was consistent between biological replicates. Furthermore, expression of exogenous UBE2E1 led to moderate decreases in MKS1 levels for both shScr and sh*Ube2e1* knock-down cells, suggesting that UBE2E1 inhibits both MKS1 levels and polyubiquitination of this protein.

To substantiate the central role of UBE2E1 in regulating MKS1 levels, we confirmed that sh*Ube2e1* cells had increased levels of polyubiquitinated cmyc-tagged MKS1 using TUBE (Tandem Ubiquitin Entity) assays. Total polyubiquitinated proteins from cell extracts were pulled-down using TUBEs bound to agarose beads, resolved by SDS-PAGE and analysed by western blotting using an anti-cmyc antibody. TUBE assays confirmed that MKS1 was polyubiquitinated (*Figure 6b*, upper panel). Treatment of pull-downs with either a pan-specific deubiquitinase (DUB) and a DUB specific for K63-linked polyubiquitination confirmed that MKS1 polyubiquitination occurred through both K63 and other ubiquitin lysine linkages such as K48 (*Figure 6b*, lower panel). Importantly, these results suggest that ubiquitination of MKS1 has dual functions in targeting this protein for degradation, as well as other regulatory functions through K63. Although UBE2E1 could be an E2 in an MKS1 degradation pathway, our data suggests that loss of UBE2E1 caused an increase in the levels of polyubiquitinated MKS1 consistent with an inhibitory function for UBE2E1 in ubiquitinating MKS1. To test this alternative hypothesis, we therefore performed in vitro ubiquitination assays in which purified MKS1 was used as a substrate of the reaction, purified UBE2E1 was the E2 and RNF34 was a possible cognate E3. We observed that UBE2E1 underwent auto-ubiquitination and that ubiquitination was enhanced but not dependent on RNF34 (*Figure 6c*, *Figure 6—figure supplement 1a*). MKS1 inhibited UBE2E1 ubiquitination, suggesting that this could be the basis for the co-dependent regulation of protein levels. This action was attenuated by addition of β-catenin, but β-catenin by itself did not inhibit

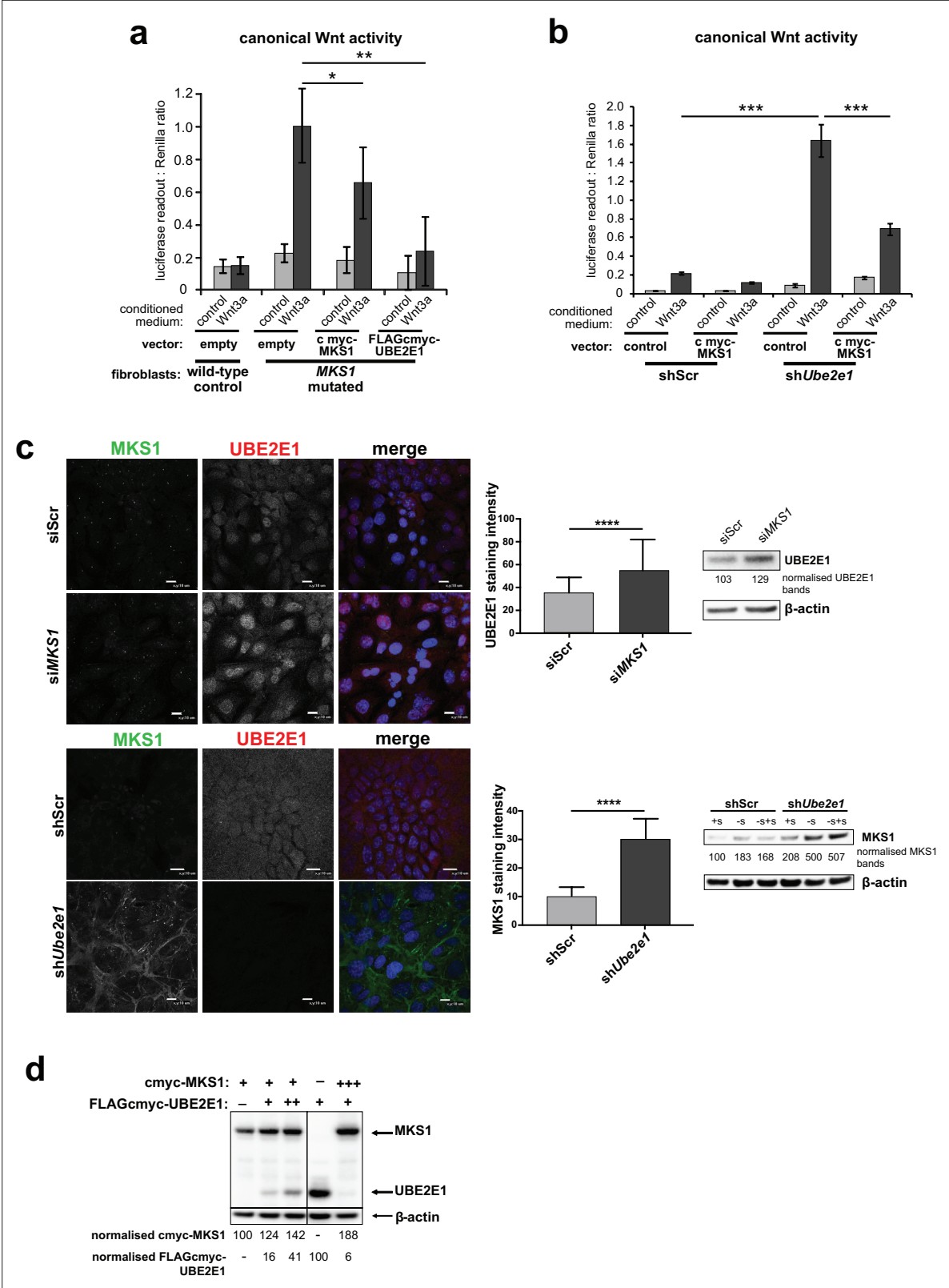

**Figure 5.** Co-dependant regulation of MKS1 and UBE2E1. (**a**) SUPER-TOPFlash assays in wild-type or *MKS1*-mutated fibroblasts, following transient co-transfection with either exogenous control, MKS1-cmyc or UBE2E1-FLAG-cmyc, and treatment with either Wnt3a or control conditioned medium. Statistical significance of the indicated pairwise comparisons with control for three independent biological replicates are shown (* p < 0.05, ** p < 0.01, *** p < 0.001, **** p < 0.0001; unpaired two-tailed Student t-test; error bars indicate s.e.m.) (**b**) SUPER-TOPFlash assays in shScr and sh*Ube2e1*

*Figure 5 continued on next page*

*Figure 5 continued*

cell-lines, following transient co-transfection with either exogenous cmyc-MKS1 or empty plasmid construct (control) and treatment with either Wnt3a or control conditioned medium, as indicated. Statistical comparisons as for (**a**. (**c**) Top panel: increased per cell staining intensity for UBE2E1 following *MKS1* siRNA knockdown Bottom panel: increased per cell staining intensity for MKS1 in *Ube2e1* mIMCD3 knockdown cells Scale bars = 10 µm. Bar graphs quantitate staining intensities for three independent biological replicates. Statistical significance of pairwise comparisons as for (**a**), error bars indicate s.e.m. Western blots (panels on right) show increased UBE2E1 protein levels for si*MKS1* knockdown cells, and increased MKS1 protein levels for sh*Ube2e1* cells. Quantitation of band intensities were normalised to β-actin loading control. (**d**) HEK293 cells were transiently transfected with control vector (-), constant (+) or high (+++) levels of cmyc-MKS1 and/or FLAG-cmyc-UBE2E1. Levels were normalised to β-actin loading control. MKS1 levels moderately decreased with increasing levels of UBE2E1, whereas high levels of MKS1 caused loss of UBE2E1.

The online version of this article includes the following source data for figure 5:

**Source data 1.** Co-dependant regulation of MKS1 and UBE2E1: full western blots.

UBE2E1 ubiquitination. However, poly-ubiquitination of β-catenin appeared to be dependent on MKS1, suggesting that UBE2E1 and MKS1 are co-regulators of β-catenin ubiquitination.

## MKS1 and UBE2E1 interact to regulate β-catenin ubiquitination

Monoubiquitination by UBE2E1 has been previously described (*Schumacher et al., 2013*) and UBE2E1 has also been shown to be an E2 ubiquitin-conjugating enzyme required for β-catenin poly-ubiquitination (*Chitalia et al., 2008*). These studies suggest that UBE2E1 has dual functions as an E2 in regulating protein function (for example, through monoubiquitination of MKS1) or targeting them for degradation (for example, polyubiquitination of β-catenin). We therefore asked the question if the co-dependent regulation of MKS1 and UBE2E1 could regulate cellular β-catenin levels. We first confirmed that MKS1 and β-catenin interact (*Figure 7*, *Figure 4—source data 1*) and that sh*Ube2e1* knock-down cells have increased levels of β-catenin (*Figure 7a*), consistent with the up-regulated canonical Wnt/β-catenin signalling that we observed in these cells (*Figure 4h*). Consistent with the mutual inhibition of MKS1 and UBE2E1, TUBE pull-down assays confirmed that levels of polyubiquitinated β-catenin increased following *MKS1* knockdown (*Figure 7b*). As expected, levels of polyubiquitinated β-catenin further increased in the presence of the catalytically-inactive dominant negative (DN) form of UBE2E1 compared to wild-type (WT) UBE2E1 (*Figure 7b*). We also observed increased levels of phosphorylated β-catenin following loss of MKS1 (*Figure 1a*, *Figure 6—figure supplement 1b*) but no effect on γ-tubulin levels (*Figure 6—figure supplement 1b*) and no consistent effect on levels of non-phosphorylated (active) β-catenin (*Figure 6—figure supplement 1b-c*). Specific localisation of phosphorylated β-catenin at the base of cilia increased following MKS1 loss (*Figure 7c*), suggesting that this is the cellular location where the phosphorylated form of β-catenin is processed by UBE2E1 for polyubiquitination. We reasoned that, in steady state conditions, UBE2E1 could mediate correct polyubiquitination levels of β-catenin, followed by subsequent targeted degradation, maintaining regulated levels of canonical Wnt signalling. In the event of high levels of the E2, caused by absence of the regulator MKS1, β-catenin is over-polyubiquitated and its levels increase leading to dysregulation of canonical Wnt signalling (*Figure 7d*). These observations indicate that correct β-catenin levels are tightly regulated at the primary cilium by a ciliary-specific E2 (UBE2E1) and a regulatory substrate-adaptor (MKS1).

## Discussion

A number of studies suggest that the primary cilium or basal body constrains canonical Wnt/β-catenin signalling activity (*Lancaster et al., 2011*; *Corbit et al., 2008*; *Simons et al., 2005*; *Abdelhamed et al., 2019*), and de-regulated, increased signalling is one of the hallmarks of the ciliopathy disease state. Canonical Wnt/β-catenin signalling is aberrantly up-regulated in several ciliopathy animal models, and, in particular, in postnatal cystic kidneys (*Kim et al., 2009*; *Lancaster et al., 2009*). We have shown previously that homozygous *Mks1*[-/-] mouse embryos also have up-regulated canonical Wnt signalling, reduced numbers of primary cilia and increased proliferation in the cerebellar vermis and kidney (*Wheway et al., 2013*). The mechanistic detail of Wnt signalling de-regulation in ciliopathies remains unclear and controversial. A key question remains whether this ciliary signalling defect is a secondary consequence of cilia loss, or if it is directly and causally related to the loss of function of specific cilium proteins. Several studies support the latter hypothesis, including one study that

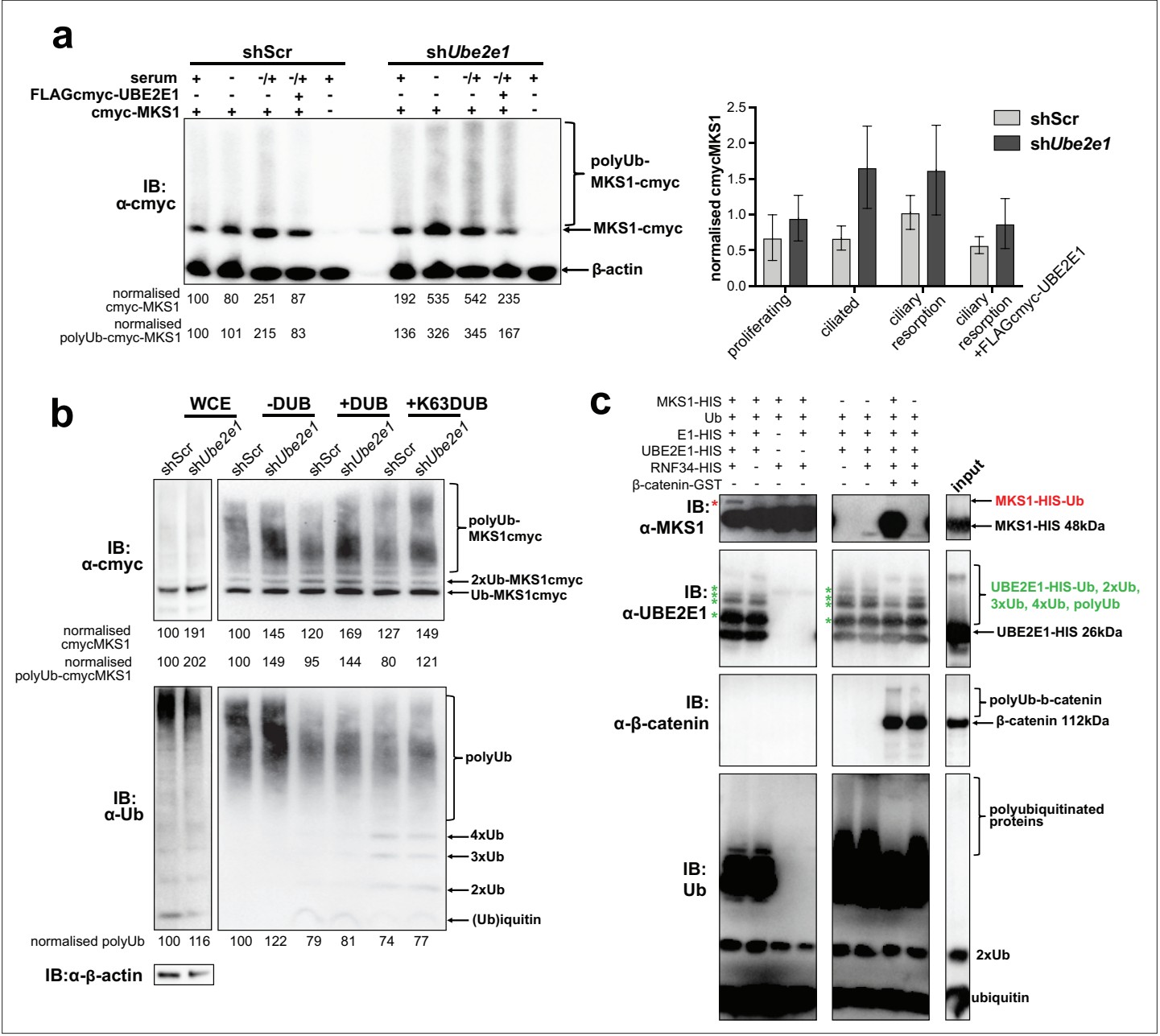

**Figure 6.** MKS1 is ubiquitinated and its ubiquitynation depends on UBE2E1. (**a**) shScr and sh*Ube2e1* mIMCD3 knockdown cells transiently transfected with cmyc-MKS1 and/or FLAG-cmyc-UBE2E1 under different conditions of ciliogenesis: proliferating cells grown in media containing serum (+); ciliated cells grown in the absence of serum (-); and cells undergoing ciliary resorption grown in the absence of serum followed by 2 hr incubation in media with serum (-/+). Increased levels of cmyc-MKS1 and smears representing poly-ubiquitinated (polyUb) cmyc-MKS1 in sh*Ube2e1* cells are indicated. Addition of exogenous FLAG-cmyc-UBE2E1 partially rescued correct MKS1 levels and ubiquitination. Normalised band intensities for the whole cmyc-MKS1 staining and only polyUb-cmyc-MKS1 are shown below the blots. Bar graph quantitates cmyc-MKS1 levels normalised to β-actin levels for three independent biological replicates. Data was analysed by two-way ANOVA followed by Tukey's multiple comparison test (statistical significance of comparison between shScr and sh*Ube2e1* is $p < 0.05$, error bars represent s.d.). (**b**) TUBE experiment confirming ubiquitination of cmyc-MKS1. Consistently increased levels of polyubiquitinated cmyc-MKS1 were observed in sh*Ube2e1* knockdown cells. Broad-range deubiquitinating enzymes (+ DUB) and K63-specific (+ K63 DUB) deubiquitinating enzyme were used to assess the type of MKS1 ubiquitination. Normalised band intensities are shown below the blots. (**c**) In vitro ubiquitination assay for MKS1-HIS, UBE2E1-HIS, RNF34-HIS, E1-HIS, Ub, and β-catenin-GST fusion proteins. The MKS1 blot shows possible mono-ubiquitination of MKS1 (red asterisk) in the presence of UBE2E1 and RNF34. Auto-ubiquitination of UBE2E1 (green asterisks indicate the addition of one, two, three, four and poly-ubiquitin chains) was inhibited by MKS1. This was further inhibited by addition of β-catenin, but β-catenin addition by itself did not affect UBE2E1 polyubiquitination. although β-catenin was polyubiquitinated by UBE2E1.

The online version of this article includes the following source data and figure supplement(s) for figure 6:

*Figure 6 continued on next page*

*Figure 6 continued*

**Source data 1.** MKS1 is ubiquitinated and its ubiquitynation depends on UBE2E1: full western blots.

**Figure supplement 1.** MKS1 is mono-ubiquitinated in presence of UBE2E1 and RNF34.

suggests that jouberin, a component of the TZ/basal body, may modulate Wnt/β-catenin signalling by facilitating nuclear translocation of β-catenin in response to Wnt stimulation (*Lancaster et al., 2009*). Regulation of Wnt signalling appears to be also mediated by a functional association of the basal body with the UPS (*Gerdes et al., 2007*), through which signalling pathway components such as β-catenin are degraded (*Aberle et al., 1997*). Early studies showed that the basal body and the proteasome can colocalise (*Fabunmi et al., 2000*; *Wigley et al., 1999*) and normal, regulated Wnt signalling has been shown to be dependent on the interaction of the basal body protein BBS4 with RPN10, a component

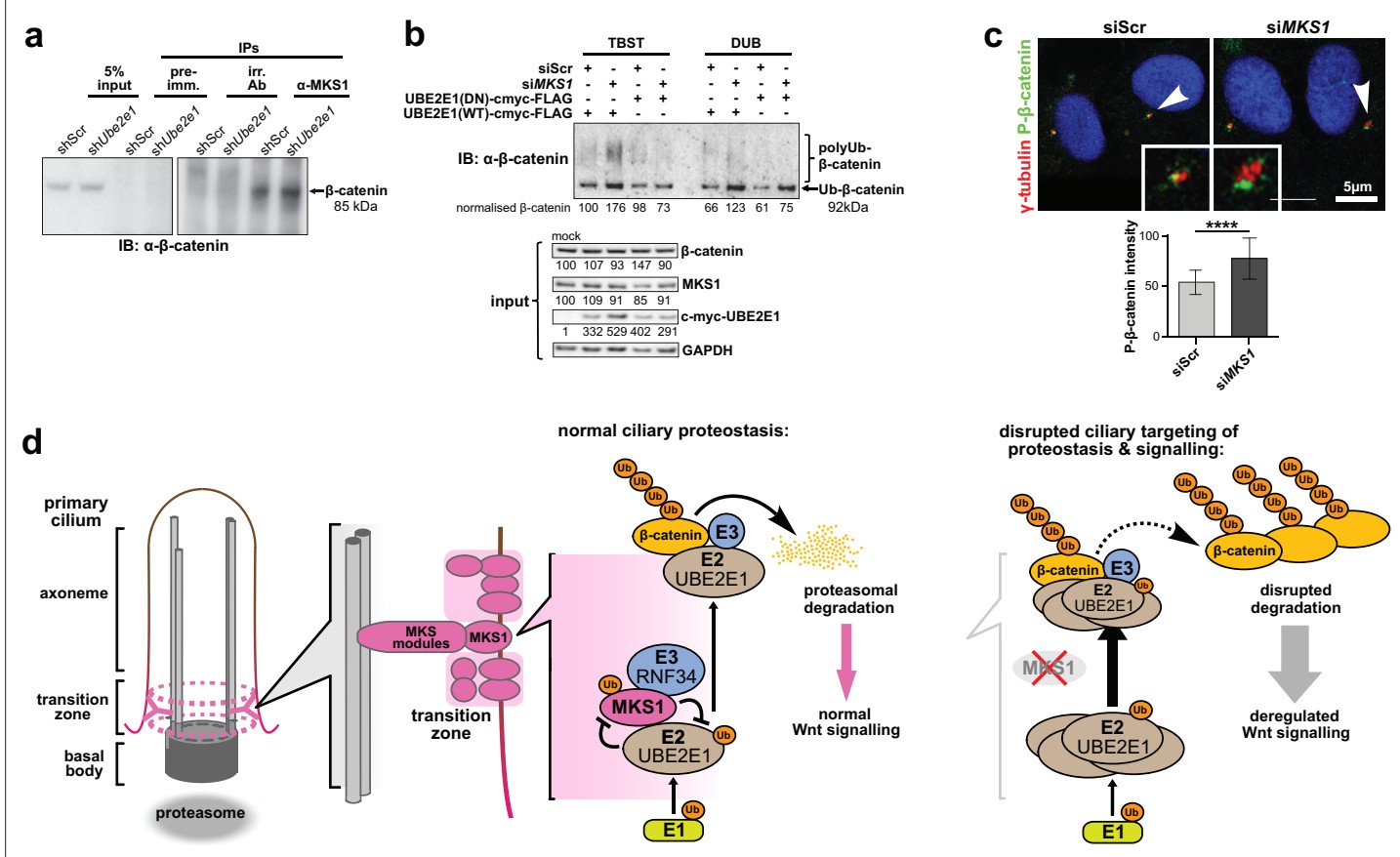

**Figure 7.** MKS1 and UBE2E1 interact to regulate β-catenin ubiquitination. (**a**) Immunoblot showing increased co-immunoprecipitation of β-catenin by anti-MKS1 in sh*Ube2e1* knockdown cells compared to shScr control cells. (**b**) TUBE pulldown followed by β-catenin immunoblotting, showing polyubiquitination of β-catenin was increased following *MKS1* knockdown and in the presence of the wild-type (WT) UBE2E1. Inactive form of UBE2E1 (DN) had not had effect on polyubiquitination of β-catenin, highlighting the importance of this UBE2E1 in β-catenin degradation. (**c**) Immunofluorescence staining of hTERT-RPE1 cells showing co-localisation of phosphorylated (**P**)-β-catenin (green) with γ-tubulin (red) at the base of cilia (arrowheads). P-β-catenin localisation significantly increased following si*MKS1* knockdown (paired two-tailed Student t-test, **** p < 0.0001 for three independent biological replicates; > 40 cells quantified per replicate). Scale bar = 5 µm. (**d**) Schematic representation of UPS regulation of MKS1 and β-catenin protein levels at the ciliary apparatus. Protein levels of MKS1 (pink) and UBE2E1 (light brown) are co-dependant through regulation at the base of the cilium. MKS1 localises to the TZ (dashed pink lines) and is mono-/bi-ubiquitinated by a complex that includes UBE2E1 and RNF34 (blue). MKS1 and UBE2E1 regulate each other, what has an effect on downstream UBE2E1 role in regulation of polyubiquitination of β-catenin (yellow). The correct regulation between these proteins facilitates normal proteasomal function and canonical Wnt signalling (small pink arrow). Both processes are de-regulated following MKS1 mutation of loss (red cross), causing aberrant accumulation of UBE2E1 and polyubiquitinated β-catenin and disrupted tethering to the ciliary apparatus.

The online version of this article includes the following source data for figure 7:

**Source data 1.** MKS1 and UBE2E1 interact to regulate β-catenin ubiquitination: full western blots.

of the proteasome (*Gerdes et al., 2007*). *Rpgrip1l⁻ᐟ⁻* knock-out mice have decreased proteasome activity, and a component of the proteasome (Psmd2), was shown to interact with Rpgrip1l (*Gerhardt et al., 2015*). Loss of RPGRIP1L does not alter the amount of MKS1 at the ciliary transition zone of mouse embryonic fibroblasts (*Wiegering et al., 2018*), but one explanation for the higher proteasome activity that we observe is that MKS1 deficiency results in an increased levels of RPGRIP1L. Our results suggest that loss or mutation of MKS1 had no consistent effect on RPGRIP1L levels (*Figure 6—figure supplement 1c*), although UBE2E1 regulation of the levels of other ciliary proteins is a mechanism that warrants further investigation. Furthermore, a number of other UPS proteins have been implicated in ciliopathies including TOPORS, an E3 ligase located at the basal body and cilium that is mutated in retinitis pigmentosa (RP) (*Chakarova et al., 2007*), and TRIM32, an E3 mutated in Bardet-Biedl syndrome (BBS) (*Chiang et al., 2006*). Interestingly, TRIM32 has been shown to interact with UBE2E1 (*Napolitano et al., 2011*). UPS components have also been shown to interact with ciliopathy proteins, such as USP9X with lebercilin (*den Hollander et al., 2007*), and the UPS was an enriched biological module that we identified in a whole genome siRNA screen of ciliogenesis (*Wheway et al., 2015*). These observations therefore support a specific role for MKS1 in UPS-mediated proteostasis and signalling regulation.

Here, we demonstrate that loss of MKS1 causes aberrant accumulation of β-catenin (*Figure 1a*) and aberrantly increased proteasome activity (*Figure 1c–d*). The increase in proteasomal activity may be a non-specific response to cellular stress in the absence of MKS1, but our discovery and validation of direct interactions of MKS1 with two proteins (UBE2E1 and RNF34) in the ubiquitination cascade suggest that loss of MKS1 causes a more specific defect. We confirmed biochemical and functional interactions of MKS1 with both UBE2E1 and RNF34 (*Figure 2c–f*, *Figure 2—figure supplement 1*), as well as their co-localisation with the basal body (*Figure 3*, *Figure 2—figure supplement 1*). Loss or dominant negative expression of UBE2E1 mimicked the cellular phenotype of *MKS1* mutants (*Figure 4a–d and g–h*), suggesting a functional interaction between MKS1 and UBE2E1 at the cilium. Using western blotting (*Figure 5d*) we substantiated an inverse correlation between MKS1 and UBE2E1 protein levels in the cell, and using in vitro ubiquitination assays (*Figure 6c*, *Figure 6—figure supplement 1a*) we show that this is due (at least in part) to ubiquitination of MKS1 by UBBE2E1. We suggest that this interaction between MKS1 and UBE2E1 plays a role in regulating Wnt signalling at the base of the primary cilium. In support of this, we demonstrated a functional interaction between UBE2E1, MKS1 and β-catenin (*Figure 7a and b*) and that phosphorylated β-catenin localised at the base of the cilium (*Figure 7c*), presumably prior to UPS processing. However, alternative interpretations of our data are that UBE2E1 could occupy an E2 binding position on MKS1 preventing MKS1 degradation, or that MKS1 binds to UBE2E1 to facilitate UBE2E1 ubiquitination and degradation.

Our data indicates that MKS1 acts as a novel substrate-adaptor that interacts with UPS components and β-catenin, thereby regulating levels of β-catenin through normal degradation during Wnt signalling. MKS1 could mediate the degradation of β-catenin by controlling the stability and the localisation of UBE2E1 at the ciliary apparatus, and perhaps ensuring the correct processing of ubiquitinated β-catenin through close proximity to the proteasome at the ciliary base. This suggestion is supported by the biochemical interaction of another ciliopathy protein, Rpgrip1l, with proteasome proteins and the discrete localisation of ubiquitin at the ciliary base (*Gerhardt et al., 2015*). Catalytically active UBE2E1 regulated ciliogenesis (*Figure 4b and d*), which implies that UBE2E1-mediated ubiquitination of substrates such as MKS1 and β-catenin is required for ciliogenesis (*Wen et al., 2010*; *Liang et al., 2008*). In addition, we show for the first time, that MKS1 is polyubiquitinated with non-degradative K63-linked chains, which have been shown to have scaffolding roles in other cell signalling networks by bridging together large signalling complexes (*Hu and Sun, 2016*). Since MKS1 contains a predicted lipid-binding B9/C2 domain, MKS1 may therefore act as a membrane anchor to ensure the spatial organisation and co-ordinated regulation of both the β-catenin destruction complex (*Corbit et al., 2008*) and UPS components at the ciliary apparatus. Loss of MKS1 would lead to the disruption of both the structure and function of the ciliary transition zone, preventing regulated ciliary signalling and β-catenin degradation (*Figure 7d*). In summary, our results indicate that the MKS1-UBE2E1 complex plays a key role in the degradation of β-catenin, which in turn facilitates correct cell function and signalling. Our data provide a mechanistic explanation for Wnt signalling defects in ciliopathies and highlights new potential targets in the UPS for therapeutic intervention.

# Materials and methods

## Key resources table

| Reagent type (species) or resource | Designation | Source or reference | Identifiers | Additional information |
|---|---|---|---|---|
| Strain, strain background (*Mus musculus*) | B6;129P2-Mks1tm1a(EUCOMM)Wtsi | Wellcome Trust Sanger Institute | EM:05429 | RRID:IMSR_EM:05429 |
| Strain, strain background (*Mus musculus*) | B6.Cg-Tg(*ACTB-Ub*G76V/*GFP*)1Dant/J | Jackson Laboratory, Maine, USA | 008111 | RRID:IMSR_JAX:008111 |
| Cell line (*Mus musculus*) | mIMCD3 | ATCC | CRL-2123 | RRID:CVCL_0429 |
| Cell line (*Homo sapiens*) | hTERT-RPE1 | ATCC | CRL-4000 | RRID:CVCL_4388 |
| Cell line (*Homo sapiens*) | ARPE-19 | ATCC | CRL-2302 | RRID:CVCL_0145 |
| Cell line (*Homo sapiens*) | HEK293 | ATCC | ACS-4500 | RRID:CVCL_4V93 |
| Cell line (*Homo sapiens*) | MKS-562 fibroblasts | *Khaddour et al., 2007* | | MKS1 compound heterozygote mutations |
| Transfected construct (*Homo sapiens*) | pCMV-cmyc-MKS1 | *Dawe et al., 2009* | | full-length cDNA (NM_017777); see Cloning, plasmid constructs and transfection |
| Transfected construct (*Homo sapiens*) | pGEX5X-1-UBE2E1 | *Hong et al., 2008* | | construct used to generate UBE2E1 protein |
| Transfected construct (*Homo sapiens*) | pCMV-UBE2E1-FLAG-cmyc | *Hong et al., 2008* | | |
| Transfected construct (*Homo sapiens*) | pCMV-UBE2E1 (DN) -FLAG-cmyc | this paper | | c.341T > A, p.C131S UBE2E1 active site dominant negative (DN) mutation; see Cloning, plasmid constructs and transfection |
| Transfected construct (*Mus musculus*) | *Ube2e1* shRNA | Origene | TR502364 | Cells selected using 0.5 µg/ml puromycin for five passages |
| Transfected construct (*Mus musculus*) | *Ube2e1* siRNA | Dharmacon ON-TARGET PLUS siRNA | L-062416-01-0005 | |
| Transfected construct (*Mus musculus*) | Mks1 siRNA | Dharmacon ON-TARGET PLUS siRNA | L-063962-01-0005 | |
| Antibody | Anti-cmyc, clone 9E10 (mouse monoclonal) | Sigma-Aldrich Co. Ltd. | M4439 | RRID:AB_439694 WB: 1:1,000 |
| Antibody | Anti-acetylated-α-tubulin, clone 6-11B-1 (mouse monoclonal) | Sigma-Aldrich Co. Ltd. | MABT868 | RRID:AB_2819178 IF: 1:1,000 |
| Antibody | Anti-HA, clone HA-7 (mouse monoclonal) | Sigma-Aldrich Co. Ltd. | H9658 | RRID:AB_260092 WB: 1:100 |
| Antibody | Anti-GFP (rabbit polyclonal) | Living Colors A.v. Peptide Antibody | 632,377 | RRID:AB_2313653 IF: 1:100 |
| Antibody | Anti-UBE2E1, clone 42/UbcH6 (mouse monoclonal) | BD Biosciences Inc | 611,218 | RRID:AB_398750 IF: 1:100 WB: 1:500 |
| Antibody | Anti-UBE2E1 (rabbit polyclonal) | Aviva Systems Biology | ARP43012_P050 | RRID:AB_2048646 IF: 1:100 WB: 1:500 |
| Antibody | Anti-γ-tubulin (rabbit polyclonal) | Sigma-Aldrich Co. Ltd. | T5192 | RRID:AB_261690 IF: 1:500 |
| Antibody | Anti-β-actin, clone AC-15 (mouse monoclonal) | Abcam Ltd. | ab6276 | RRID:AB_2223210 WB: 1:5,000 |
| Antibody | Anti-cyclin D1, clone A-12 (mouse monoclonal) | Santa Cruz Biotechnology Inc | sc-8396 | RRID:AB_627344 WB: 1:1,000 |
| Antibody | Anti-phospho-β-catenin (rabbit polyclonal) | Cell Signalling Technology Inc | 9,561 | RRID:AB_331729 WB: 1:1,000 IF: 1:100 |
| Antibody | Anti-β-catenin, clone D10A8 (rabbit monoclonal) | Cell Signalling Technology Inc | 8,480 | RRID:AB_2798305 WB: 1:1,000 IF: 1:100 |

*Continued on next page*

*Continued*

| Reagent type (species) or resource | Designation | Source or reference | Identifiers | Additional information |
|---|---|---|---|---|
| Antibody | Anti-mono- and polyubiquitinylated conjugates, clone FK2 (mouse monoclonal) | Enzo Life Sciences, Inc | ENZ-ABS840 | RRID:AB_10541840<br>WB: 1:1,000 |
| Antibody | Anti-20S proteasome α7 subunit, clone MCP72 (rabbit monoclonal) | Enzo Life Sciences Inc | BML-PW8110 | RRID:AB_10538395<br>WB: 1:1,000<br>IF: 1:100 |
| Antibody | Anti-MKS1 (rabbit polyclonal) | *Dawe et al., 2007*; *Näthke et al., 1996* | | WB: 1:500<br>IF: 1:100 |
| Antibody | Anti MKS1 (rabbit polyclonal) | Proteintech | 16206–1-AP | RRID:AB_10637856<br>WB: 1:500<br>IF: 1:100 |
| Antibody | Anti-ubiquitin, clone P4D1 (mouse monoclonal) | Santa Cruz Biotechnology, Inc | sc-8017 | RRID:AB_2762364 |
| Peptide, recombinant protein | MKS1-HIS | Proteintech Group, Inc | Ag9504 | |
| Peptide, recombinant protein | UBE2E1-HIS | Enzo Life Sciences, Inc | UW8710 | |
| Peptide, recombinant protein | RNF34-HIS | Novus Biologicals | NBP2-23440 | |
| Peptide, recombinant protein | β-catenin-GST | Abcam | Ab63175 | |
| Commercial assay or kit | 20 S fluorophore substrate Suc-LLVY-AMC | Enzo Life Sciences Inc | BML-P802-0005 | |
| Commercial assay or kit | Dual-Luciferase Reporter Assay system | Promega Corp. | E1910 | |
| Commercial assay or kit | Ubiquitination kit | Enzo Life Sciences, Inc | BML-UW0400 | |
| Commercial assay or kit | TUBE assays | LifeSensors, Malvern, PA, USA | UM-402 | |
| Chemical compound, drug | MG-132 | Sigma-Aldrich Co. Ltd. | C2211 | treatment at 10 µM for 3 hr |
| Software, algorithm | Prism7 | GraphPad Software Inc | | |

## Informed consent for use of patients in research

Informed consent was obtained from all participating families or patients, with studies approved by the Leeds (East) Research Ethics Committee (REC no. 08 /H1306/85) on 4th July 2008.

## Animals

The animal studies described in this paper were carried out under the guidance issued by the Medical Research Council in *Responsibility in the Use of Animals for Medical Research* (July 1993) in accordance with UK Home Office regulations under the Project Licence no. PPL40/3349. B6;129P2-Mks1$^{tm1a(EUCOMM)Wtsi}$ heterozygous knock-out mice were derived from a line generated by the Wellcome Trust Sanger Institute and made available from MRC Harwell through the European Mutant Mouse Archive http://www.emmanet.org/ (strain number EM:05429). The *Ub$^{G76V}$-GFP* line (25) B6.Cg-Tg(*ACTB-Ub*$^{*G76V}$/GFP)1$^{Dant/J}$ (strain number 008111) was obtained from the Jackson Laboratory, Maine, USA. Genotyping was done by multiplex PCR on DNA extracted from tail tips or the yolk sac of E11.5-E15.5 embryos, or ear biopsies of adult mice. Primer sequences: exon 2 F: TGGGGAAGGACCTCATAGACT, exon 4 R: CGCCAGAATTCTCCAGTTTC, exon 4 F: AGCGTGGTTGTTCTTGATGA, exon 6 R: GGATTCCGCACTGAGACAAC, exon 16 F: AACCGGCGAATCTTCACTTA, exon 18 R: GGGGCTCACAAGGTCCTG. Proteasome inhibition treatment of *Mks1* x *Ub$^{G76V}$-GFP* mice using MG-262 was carried out as previously described (*Lindsten et al., 2003*).

## Preparation of tissue sections

Mouse embryos or tissue for IF staining were lightly fixed in 0.4% paraformaldehyde, soaked in 30% sucrose/PBS, frozen in OCT embedding medium and cut into 5 µm sections on a cryostat. Fresh-frozen sections were left unfixed and processed for immunofluorescent staining by standard techniques.

## Cells

Mouse inner medullary collecting duct (mIMCD3), human retinal pigment epithelium cells immortalised with human telomerase reverse transcriptase (hTERT-RPE1) and immortalised adult retinal pigment epithelium (ARPE19) cells were grown in Dulbecco's minimum essential medium (DMEM)/Ham's F12 supplemented with 10% foetal calf serum at 37 °C/5% $CO_2$. Human embryonic kidney (HEK293) cells were cultured in DMEM with 10% foetal calf serum at 37 °C/5% $CO_2$. Cell-lines were maintained by weekly passaging under standard conditions and tested every 3 months for mycoplasma. Cell lines were sourced from American Type Culture Collection (ATCC) and used between passages 15–25. Cell-lines have been previously verified using arrayCGH and RNA-sequencing (*Wheway et al., 2015*) (Short Read Archive accession numbers SRX1411364, SRX1353143, SRX1411453, and SRX1411451). The derivation and culture of mouse embryonic fibroblasts (MEFs) has been described previously (*Xu, 2001*). MEFs were grown in DMEM/Ham's F12 supplemented with 10% foetal calf serum and 1% penicillin streptomycin at 37 °C/5% $CO_2$. Fibroblasts from a normal undiseased control, a patient (MKS-562) with a compound heterozygous *MKS1* mutation, and a female patient with a homozygous *ASPM* mutation, were immortalised following transduction with an amphotropic retrovirus encoding the hTERT catalytic subunit of human telomerase, and maintained in Fibroblast Growth Medium (Genlantis Inc San Diego, CA) supplemented with 0.2 mg/ml geneticin. Patient MKS-562, a compound heterozygote for the *MKS1* mutations [c.472C > T]+[IVS15-7_35del29] causing the predicted nonsense and splice-site mutations [p.R158*]+[p.P470*fs*562], has been described previously (*Khaddour et al., 2007*). Proteasome inhibition treatment was carried out using 10 µM final concentration of the inhibitor dissolved in DMSO for 16 hr (unless otherwise stated). DMSO was used as the vehicle-only negative control.

## Cloning, plasmid constructs, and transfection

Human *MKS1* was cloned into the pCMV-cmyc vector as described previously (*Dawe et al., 2009*). The pGEX5X-1-UBE2E1 and pCMV-UBE2E1-FLAG-cmyc constructs have been described previously (*Hong et al., 2008*). The c.341T > A, p.C131S active site dominant negative (DN) missense mutation was introduced into pCMV-UBE2E1-FLAG-cmyc using the QuickChange mutagenesis kit (Stratagene Inc) and verified by DNA sequencing. For transfection with plasmids, cells at 80% confluency were transfected using Lipofectamine 2000 (Invitrogen Inc) according to the manufacturer's instructions and as described previously (*Dawe et al., 2009*). Cells transfected with plasmids expressing *Ube2e1* shRNA (Origene) were selected for using 0.5 µg/ml puromycin for five passages. Transfection with Dharmacon ON-TARGET PLUS siRNAs was carried out using Lipofectamine RNAiMAX according to the manufacturer's instructions and as described previously (*Dawe et al., 2009*). To assess co-dependency of protein levels, 1 µg of cmyc-MKS1 was co-transfected with 1, 2.5, and 5 µg of FLAG-cmyc-UBE2E1. To investigate if an increased amount of MKS1 would have an effect on UBE2E1 levels, 3 µg of cmyc-MKS1 were co-transfected with 1 µg FLAG-cmyc-UBE2E1. After 24 hr incubation with transfection complexes, cells were treated with 100 µg/ml cycloheximide for 4 hr. Ubiquitination of cmyc-MKS1 in mIMCD3 cells was assessed after treatment with proteasome inhibitor (MG-132 at 10 µM) for 3 hr.

## Antibodies

The following primary antibodies were used: mouse anti-cmyc clone 9E10, mouse anti-acetylated-α-tubulin clone 6-11B-1, mouse anti-HA (Sigma-Aldrich Co. Ltd.), rabbit anti-GFP ('Living Colors A.v. Peptide Antibody') and mouse anti-UBE2E1 (BD Biosciences Inc); rabbit-anti-γ-tubulin and mouse anti-β-actin clone AC-15 (Abcam Ltd.); mouse anti-cyclin D1 clone A-12 (Santa Cruz Biotechnology Inc); rabbit anti-phospho-β-catenin and rabbit anti-β-catenin (Cell Signalling Technology Inc); and mouse anti-mono- and polyubiquitinylated conjugates clone FK2 and rabbit anti-20S proteasome α7 subunit (Enzo Life Sciences Inc). Rabbit anti-MKS1 has been described previously (*Dawe et al., 2007*; *Näthke et al., 1996*). Secondary antibodies were AlexaFluor488-, and AlexaFluor568-conjugated

goat anti-mouse IgG and goat anti-rabbit IgG (Molecular Probes Inc) and HRP-conjugated goat anti-mouse immunoglobulins and goat anti-rabbit immunoglobulins (Dako Inc).

## Immunofluorescence and confocal microscopy

Cells were seeded at $1.5 \times 10^5$ cells/well on glass coverslips in six-well plates, 24 hr before transfection and 48–96 hr before fixation. Cells were fixed in ice-cold methanol (5 min at 4 °C) or 2% paraformaldehyde (20 min at room temperature). Permeabilisation, blocking methods and immunofluorescence staining were essentially as described previously (*Valente et al., 2010*). Confocal images were obtained using a Nikon Eclipse TE2000-E system, controlled and processed by EZ-C1 3.50 (Nikon Inc) software. Images were assembled using Adobe Photoshop CS3 and Adobe Illustrator CS2.

## Yeast 2-hybrid screening

The B9/C2 domain of human *MKS1* (amino acids 144–470; *Figure 4a*) was cloned into the Gal4 vector pB27 and screened against a human fetal brain RP1 prey cDNA library. Yeast-2-hybrid screens were performed by Hybrigenics SA as described previously (*Dawe et al., 2009*). Confirmatory '1-to-1' pairwise assays for selected interactants were performed with the MatchMaker Two-Hybrid System 3 (Clontech Inc).

## GST fusion protein purification

GST-UBE2E1 fusion protein was prepared essentially as described previously (*Hong et al., 2008*), with protein expression induced at 20 °C using 0.2 mM IPTG for 4 hr.

## Proteasome activity assays

Crude proteasomal fractions were prepared from cells (*Hoffman et al., 1992*) and incubated with the 20 S fluorophore substrate Suc-LLVY-AMC (Enzo Life Sciences Inc). Fluorescence of each proteasomal preparation was measured on a Mithras LB940 (Berthold Technologies Inc) fluorimeter and adjusted against a calibration factor calculated from a standard curve to give activity measurements in pmol AMC release/μg cell lysate/hour. Treatment of cells with 10 μM of the proteasome inhibitors MG-132, MG-262 or c-lactacystin-β-lactone were positive controls for the assay. Results reported are from at least five independent biological replicates.

## Canonical Wnt activity (SUPER-TOPFlash) luciferase assays

For luciferase assays of canonical Wnt activity, we grew cells in 12-well plates and co-transfected with 0.5 μg SUPER-TOPFlash firefly luciferase construct (*Veeman et al., 2003*) (or FOPFlash, as a negative control); 0.5 μg of expression constructs (pCMV-cmyc-MKS1, or empty pCMV-cmyc vector); and 0.05 μg of pRL-TK (Promega Corp; *Renilla* luciferase construct used as an internal control reporter). We obtained Wnt3a- or Wnt5a-conditioned media from stably-transfected L cells with Wnt3a or Wnt5a expression vectors (ATCC). Control media was from untransfected L cells. Activities from firefly and *Renilla* luciferases were assayed with the Dual-Luciferase Reporter Assay system (Promega Corp.) on a Mithras LB940 (Berthold Technologies Inc) fluorimeter. Minimal responses were noted with co-expression of the FOP Flash negative control reporter construct. Raw readings were normalised with *Renilla* luciferase values. Results reported are from at least four independent biological replicates.

## Purification of UBE2E1 protein

UBE2E1-FLAGcmyc was transfected into HEK293T cells using Lipofectamine 2000 (ThermoFisher Scientific inc) Cells were incubated with transfection complexes for 3 hr, and changed to normal growing medium for further 16 hr incubation. Cells were then incubated with 10 μM MG-132 for 5 hr and whole cell extracts (WCE) prepared as described previously (*Johnson et al., 2001*). Protein lysate was incubated with ANTI-FLAG M2 affinity gel (Sigma-Aldrich Co. LLC) and purified UBE2E1-FLAGcmyc was eluted from the beads following the manufacturer's instructions.

## In vitro ubiquitination assays

To assess in vitro ubiquitination, we used a ubiquitination kit (Enzo Life Sciences, Inc) according to the manufacturer's protocol, supplemented with MKS1-HIS (Proteintech Group, Inc), UBE2E1-HIS (Enzo Life Sciences, Inc), RNF34-HIS (Novus Biologicals) and β-catenin-GST (Novus Biologicals) fusion

proteins in a total volume of 30 µl. Samples were incubated for 1.5 hr at 37 °C followed by SDS-PAGE and western blotting.

## TUBE assays

Agarose-bound TUBE assays were used as recommended by the manufacturer (LifeSensors, Malvern, PA, USA). mIMCD3 cells were transiently transfected with cmyc-MKS1 and treated with proteasome inhibitor (MG-132 at 10 µM) for 2 hr before harvesting. Lysis buffer was based on RIPA supplemented with 50 mM Tris-HCl pH7.5, 0.15 M NaCl, 1 mM EDTA, 1% NP40, 10% glycerol, DUB inhibitors (50 µM PR619 and 5 mM 1,10-phenanthroline) and protease inhibitors. 5 µM BRISC was used as K63 deubiquitinating enzyme. In short, cells were harvested after incubation with proteasome inhibitor and proteins were extracted using TUBE lysis buffer following standard procedures. Protein concentration was measured using Lowry assay and about 750 µg of protein was used in the pull down. Cell lysates were incubated with uncoupled agarose beads to remove unspecific binding proteins and were subsequently incubated with equilibrated 40 µl TUBE-agarose beads for 2 hr at 4 °C on a rocker. Beads were spun down, washed, eluted and neutralised. Samples were then split into three for incubation with TBST, DUB and BRISC for 1 hr at 37 °C. Samples were run on SDS-PAGE followed by western blotting using standard protocols. Membranes were blotted with mouse anti-cmyc (clone 9E10, Sigma-Aldrich Co. Ltd.), Ub-HRP (P4D1, Santa Cruz Biotechnology, Inc) and rabbit anti-β-catenin (Cell Signalling Technology Inc).

## Co-Immunoprecipitation and mass spectrometry

Whole cell extracts (WCE) were prepared and co-IP performed essentially as described previously (*Johnson et al., 2001*). Co-IPs used either 5 µg affinity-purified mouse monoclonals (MAbs), or 5–10 µg purified IgG fractions from rabbit polyclonal antisera, coupled to protein G- and/or protein A-sepharose beads (GE Healthcare UK Ltd.). Proteins were eluted from beads with 0.2 M glycine HCl pH2.5. Samples were neutralised by addition of 0.1 volume 1 M Tris HCl ph8.5. After elution, proteins were precipitated with chloroform and methanol and subjected to in-solution tryptic cleavage as described previously (*Gloeckner et al., 2009*). LC-MS/MS analysis was performed on Ultimate3000 nano RSLC systems (Thermo Scientific) coupled to a Orbitrap Fusion Tribrid mass spectrometer (Thermo Scientific) by a nano spray ion source (*Boldt et al., 2016*). Mascot (Matrix Science, Version 2.5.1) was used to search the raw spectra against the human SwissProt database for identification of proteins. The Mascot results were verified by Scaffold (version Scaffold_4.8.8, Proteome Software Inc, Portland, OR, USA) to validate MS/MS-based peptide and protein identifications.

## Western blotting

Soluble protein was analysed by SDS-PAGE using 4–12% Bis-Tris acrylamide gradient gels and western blotting was performed according to standard protocols using either rabbit polyclonal antisera (final dilutions of x200-1000) or MAbs (x1000-5000). Appropriate HRP-conjugated secondary antibodies (Dako Inc) were used (final dilutions of x10000-25000) for detection by the enhanced chemiluminescence 'Femto West' western blotting detection system (Pierce Inc). Chemiluminescence was detected using a BioRad ChemiDoc MP Imaging System and Image Lab software. Volumetric analysis of immunoblot bands was performed using Image Lab software (Bio Rad). Full blots are shown in the source data files associated with each figure and figure supplement, as appropriate.

## Statistical analyses

Normal distribution of data (for SUPER-TOPFlash assays, proteasome activity assays, cilia length measurements) was confirmed using the Kolmogorov-Smirnov test (GraphPad Software). Paired or unparied comparisons were analysed with either Student's two-tailed t-test, $\chi^2$ tests or other tests as detailed in figure legends as appropriate using InStat (GraphPad Software). Results reported are from at least three independent biological replicates.

## Acknowledgements

This paper is dedicated to the memory of P Robinson, our valued collaborator, colleague and friend. We are very grateful to D Evans, J Bilton, C McCartney and M Reay for technical support. We thank E Pitt for hTERT immortalization of MKS1 patient fibroblasts. We thank A Monk, K Passam and T

Simpson of Nikon UK Ltd. for technical support and advice on confocal microscopy. We are grateful to R T Moon (University of Washington) for the SUPER-TOPFlash and FOPFlash constructs, and S Kang (Korea University) for the UBE2E1 constructs. The anti-MKS1 antibody was a gift from N Katsanis, Duke University Medical Center. We acknowledge funding from the UK Medical Research Council (Doctoral Training Award for GW, project grant MR/M000532/1 to CAJ) The research also received funding from the European Community's Seventh Framework Programme FP7/2009 under grant agreement no: 241,955 SYSCILIA. KS was funded by Wellcome Trust Institutional Strategic Support Funding to University of Leeds (105615/Z/14/Z) and GW was funded by a Wellcome Trust Seed Award in Science (204378/Z/16/Z).

---

## Additional information

### Funding

| Funder | Grant reference number | Author |
|---|---|---|
| Medical Research Council | MR/M000532/1 | Colin A Johnson |
| European Community's Seventh Framework Programme FP7/2009 Health | 241955 SYSCILIA | Colin A Johnson |
| Wellcome Trust | 204378/Z/16/Z | Gabrielle Wheway |
| Wellcome Trust | 105615/Z/14/Z | Katarzyna Szymanska |

The funders had no role in study design, data collection and interpretation, or the decision to submit the work for publication.

### Author contributions

Katarzyna Szymanska, Gabrielle Wheway, Conceptualization, Data curation, Formal analysis, Investigation, Methodology, Project administration, Validation, Visualization, Writing – original draft, Writing – review and editing; Karsten Boldt, Data curation, Formal analysis, Investigation, Writing - original draft, Writing – review and editing; Clare V Logan, Matthew Adams, Data curation, Formal analysis, Investigation, Writing – review and editing; Philip A Robinson, Conceptualization, Formal analysis, Methodology, Writing – review and editing, Software; Marius Ueffing, Data curation, Funding acquisition, Methodology, Writing - original draft, Software, Writing – review and editing; Elton Zeqiraj, Formal analysis, Methodology, Writing – review and editing, Software, Writing – review and editing; Colin A Johnson, Conceptualization, Data curation, Formal analysis, Funding acquisition, Investigation, Methodology, Project administration, Writing – review and editing, Software, Visualization, Writing – original draft, Writing – review and editing

### Author ORCIDs

Katarzyna Szymanska http://orcid.org/0000-0001-7736-5225
Marius Ueffing http://orcid.org/0000-0002-9045-182X
Elton Zeqiraj http://orcid.org/0000-0003-0239-5926
Colin A Johnson http://orcid.org/0000-0002-2979-8234

### Ethics

Informed consent was obtained from all participating families or patients, with studies approved by the Leeds (East) Research Ethics Committee (REC no. 08/H1306/85) on 4th July 2008.
The animal studies described in this paper were carried out under the guidance issued by the Medical Research Council in Responsibility in the Use of Animals for Medical Research (July 1993) in accordance with UK Home Office regulations under the Project Licence no. PPL40/3349.

### Decision letter and Author response

Decision letter https://doi.org/10.7554/eLife.57593.sa1
Author response https://doi.org/10.7554/eLife.57593.sa2

## Additional files

### Supplementary files
• Transparent reporting form
• Source code 1. Source data files for gels and blots displayed in Figures 1-7 & figure supplements.

### Data availability
Data generated or analysed during this study are included in the manuscript and supporting files. Imaging data for gels and blots is collated as both original files of the full unedited files, and figures with the uncropped gels or blots with the relevant bands highlighted. Full, uncropped western blots are provided in figure supplements, as appropriate for all figures. Source data files are also included for Figure 4e-f, and for all gels and blots displayed in Figures 1-7 (apart from Figure 1e, Figure 1-figure supplement 1 panel e for beta-actin western, Figure 4-figure supplement 1 panel a). Supplementary data to support Figure 4e-f and Figure 4-figure supplement 1b is available from University of Leeds at https://doi.org/10.5518/814.

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
