## [Editor Report]

It has long been recognized that ciliary dysfunction leads to increased canonical Wnt signaling but the mechanism has been elusive. Your work connecting β-catenin stability to Mks1 through Ube2e1 is an important advance in understanding this mechanism. I am certain that your work will stimulate more effort in this important area.

---

## [Decision Letter]

**Decision letter after peer review:**

Thank you for submitting your article "Regulation of canonical Wnt signalling by the ciliopathy protein MKS1 and the E2 ubiquitin-conjugating enzyme UBE2E1" for consideration by *eLife*. Your article has been reviewed by 3 peer reviewers, including Gregory J Pazour as Reviewing Editor and Reviewer #1, and the evaluation has been overseen by Piali Sengupta as the Senior Editor. The following individual involved in review of your submission has agreed to reveal their identity: Christoph Gerhardt (Reviewer #2).

The reviewers have discussed the reviews with one another and the Reviewing Editor has drafted this decision to help you prepare a revised submission.

Summary:

This manuscript from Szymanska et al., explored MKS1 regulation of canonical Wnt signaling and uncovered a role for the E2 ubiquitin conjugating enzyme Ube2e1 in the degradation of phosphorylated β catenin. This paper addresses an important question in the field as to why ciliary dysfunction causes increased canonical Wnt signaling. The data convincing shows that the loss of MKS1 elevates canonical Wnt signaling. They then identify the E2 ligase Ube2e1 as an Mks1 binding protein by yeast two-hybrid and explore the role of Ube2e1 and Mks1 in regulating β catenin levels. The supporting data has significant problems that will need to be addressed before publication. First, many of the conclusions depend on small differences in band intensity. These differences are often not quantitated or if they are quantitated, it is not clear that the experiments were performed more than once. Statistical analysis also needs to be improved. Many experiments lack statistical tests; others use t-tests to compare multiple variables without any correction for multiple analysis. Secondly, as described below, the quality of several experiments needs to be improved if they are to be used to support the conclusions of the paper.

Essential revisions:

1. Data analysis is lacking. A number of figures lack quantitative analysis. Most lack details about the statistical analysis performed. It is not clear what "#" means in the graphs. It appears that a simple t-test were used for multiple comparisons without correction. Figure legends need to indicate the n of the experiment and the statistical test used.

2. Why would increased proteasomal activity (Figure 1C) cause GFP to accumulate in the UbG76V line? It seems that increased proteasomal activity would decrease (or not change) GFP accumulation.

3. GST pulldown in Figure 2D is not convincing. The background in the control lane is darker than the experimental lane and could obscure a band of the intensity seen in the GST-UBE2E1 pulldown.

4. Figure 3b: There is little or no difference between the "no colocalization" and "colocalization" images. There may be a quantitative difference in the centrosomal staining seen in the panels in 3a but this would need to be measured.

5. Figure 5c. Western blots need to be added. Why is the level of nuclear Ube2e1 relevant to the work?

6. The conclusion that increased expression of UBE2E1 decreases MKS1 levels (5d) is not convincing.

7. The details of what was done in Figure 6b are lacking but a number of points are confusing. Why doesn't the band marked "MKS1cmyc-Ub"show up in the Ub western blot? Would we expect all ubiquitinated proteins to be detected in the Ub blot? If so, then this blot is not evidence of K63 polyubiquitination of the MKS1. The differences between the lanes in the c-myc blot are not convincing. I would expected the DUBs to reduce the extent of polyubiquitination, which should have brought the smears of poly-Ub-MKS1cmyc down the gel, but this is not seen.

8. Figure 6c is problematic. First, both MKS1-GST and Ub-GST are used and a GST western blot is used as evidence of Ub ligation onto MKS1. Ub needs a different tag from its substrate to be convincing. Second, the patterns of the proteins in the blots do not match the guide at the top. Rnf34-His is found in lanes where it was not added. UbE2E1-Flag is only seen in the one of the lanes where it is supposed to be present. This seems to be an unduly complicated figure with UbE2E1-Flag and UbE2E1-His showing different results. There is no explanation of this in the figure legend or text.

9. Figure 6e is too low quality to be interpreted.

10. The staining of the neuroepithelium (S2) with the anti-acetylated tubulin antibody shows decreased cytoplasmic staining and no evidence of ciliary label. Better cilia markers, higher magnification and quantification are needed to make this point. This is also a problem in S1D where the MKS mutant cells have strong cytoplasmic label. It is unclear what the arrow is pointing at, but there is no evidence that it is a cilium.

11. It was published that phospho-β-catenin (S33/S37/T41) is degraded by the proteasome at the base of cilia (Corbit et al., 2008; Gerhardt et al., 2015). Szymanska et al., showed that the amount of phospho-β-catenin (S33/S37/T41) is increased by MKS1 deficiency, although proteasome activity is enhanced in MKS1 mutant cells. How can this discrepancy be explained? The authors did not provide a sufficient explanation. An explanation could be that MKS1 affects Wnt signaling and proteasome activity in two different ways. But this needs to be shown. As they mentioned in the discussion, another transition zone protein, RPGRIP1L, positively regulates proteasome activity (Gerhardt et al., 2015). Previously, it was shown that the loss of RPGRIP1L does not alter the amount of MKS1 at the ciliary transition zone of mouse embryonic fibroblasts (Wiegering et al., 2018) but maybe MKS1 deficiency results in an increased amount of RPGRIP1L at the transition zone of human and mouse fibroblasts leading to the higher proteasome activity the authors observed. Or mutations in MKS1 change the ciliary amount of BBS4 and/or OFD1 which also govern proteasome activity (Liu et al., 2014).

12. Furthermore, some examinations should be performed in another way to allow a more precise interpretation. For example, the authors quantified phospho-β-catenin (S33/S37/T41) and β-catenin. In MKS1 deficient cells, the amount of both proteins is higher. In my opinion, an important control measurement is missing. The authors should additionally measure the amount of non-phospho-β-catenin (S33/S37/T41). Since Szymanska et al., suggest that MKS1 and UBE2E1 colocalize at the ciliary base, they should also measure the amount of hosphor-β-catenin (S33/S37/T41) not only by Western blotting (overall cellular amount) but also at the ciliary base by fluorescence intensity measurements in MKS1 deficient human or mouse fibroblasts. This kind of measurement was performed in hTERT-RPE1 cells which were treated with siRNA against MKS1 (Figure 6f) but in these cells the amount of γ-tubulin seems to be increased. Maybe, the increased amount of hosphor-β-catenin (S33/S37/T41) is caused by the higher amount of γ-tubulin. It cannot be excluded that several cilia overlap because this image lacks ciliary axoneme staining. In regard to the colocalization of MKS1 and UBE2E1, the authors should perform an analysis such as FRET or in situ PLA which confirms that both proteins localize close enough together to interact at the ciliary base.

13. The proteasome activity assay uses the 20S fluorophore substrate Suc-LLVY-AMC to measure activity. The general view is that the proteasome degrades this substrate independently of ubiquitination (Gan et al., 2019). Despite of this, Szymanska et al., observe an enhanced proteasome activity in cells deficient for UBE2E1, an E2 ubiquitin-conjugation enzyme. This data should be carefully controlled in order to test whether the proteasome activity assay results are reliable. It would be helpful to check whether proteasome activity assays in which other substrates are used that are degraded by the proteasome without ubiquitination provide the same results. If this is the case, UBE2E1 may indirectly regulate proteasome activity by taking part in the ubiquitination of a protein that regulates proteasome activity.

14. In human there are approximately 40 E2 enzymes, therefore each E2 is likely to act with multiple E3s (there are over a 1000) with E3s likely to be the main driver of target specificity. Authors have not controlled their experiments, or at least have not showed it in the manuscript, with any other E2s. Consequently, there is a real risk that the findings simply confirm that cilia and MKS1 are regulated by UPS rather than that UBE2E1 is a specific regulator in this system.

15. With regard to immunoprecipitations and ubiquitination assays. Firstly inputs (for each analyzed protein in each sample) and/or loading controls are not always shown (e.g. Figure 2) which makes interpretation of results difficult.

16. Details are lacking about the conditions of the IP experiments used to detect ubiquitinated proteins (6b,d,e). If these were performed under native conditions, the ubiquitin being detected by western blot could be on any protein that complexes with the target of the precipitation. E2 ligases may have a large number of interactors.

17. As far as the UBE2E1 knockdown model. In the MS analysis in Figure 3, I am quite surprised that there is still such a strong recovery of UBE2E1 peptides from UBE2E1 silenced line. Is this not an impediment to this model? Does this not suggest that this cell line still has sufficient, functioning UBE2E1? Also again, it is hard to estimate if the MKS1 specific observations would not be replicated by silencing other E2s.

18. Lack of inputs in Figure 6a,b,d and e make interpretation of the critical claims from this manuscript difficult. If there is an effect of UBE2E1 on MKS1 total protein levels inputs of UBE2E1, MKS1 and loading controls for all samples in 6a 6b and 6d should clarify this. Currently, the only demonstration of the claimed protein increase is in Figure 5d and it is not repeated 3 times to allow for statistical analysis. Furthermore, it looks very marginal (numbers are not greatly affected either 113-107-96 with appreciation that WB is not really a quantitative technique) and unlikely to be significant. It is possible that even if direct, changes caused by UBE2E1 to MKS1and vice versa are completely irrespective of protein levels or stability and affect localization and function through chains other than K48.

19. Authors should clarify statements regarding degradation and UBE2E1 dual function, as alternative explanations have not been ruled out. UBE2E1 action on MKS1 could be indirect. The increase in b-catenin levels in Figure 6d is very hard to interpret without loading controls.

20. There are multiple alternative explanations for the observations presented in this manuscript, which renders figure 7 a bit uncertain. For example, UBE2E1 could be involved in ubiquitination of an MKS1 targeting E3 or DUB and that could be the reason for the observed changes. It would be more forthcoming to highlight this in the manuscript.

References:

Corbit, K., Shyer, A., Dowdle, W., Gaulden, J., Singla, V., Chen, M., Chuang, P., Reiter, J., 2008. Kif3a constrains β-catenin-dependent Wnt signalling through dual ciliary and non-ciliary mechanisms. Nat. Cell Biol. 10, 70-76.

Gan, J., Leestemaker, Y., Sapmaz, A., Ovaa, H., 2019. Highlighting the Proteasome: Using Fluorescence to Visualize Proteasome Activity and Distribution. Front. Mol. Biosci. 6, 14.

Gerhardt, C., Lier, J., Burmühl, S., Struchtrup, A., Deutschmann, K., Vetter, M., Leu, T., Reeg, S., Grune, T., Rüther, U., 2015. The transition zone protein Rpgrip1l regulates proteasomal activity at the primary cilium. J. Cell Biol. 210, 115-133.

Liu, Y., Tsai, I., Morleo, M., Oh, E., Leitch, C., Massa, F., Lee, B., Parker, D., Finley, D., Zaghloul, N., Franco, B., Katsanis, N., 2014. Ciliopathy proteins regulate paracrine signaling by modulating proteasomal degradation of mediators. J. Clin. Invest. 124, 2059-2070.

Wheway, G., Abdelhamed, Z., Natarajan, S., Toomes, C., Inglehearn, C., Johnson, C., 2013. Aberrant Wnt signalling and cellular over-proliferation in a novel mouse model of Meckel-Gruber syndrome.. Dev. Biol. 377, 55-66.

Wiegering, A., Dildrop, R., Kalfhues, L., Spychala, A., Kuschel, S., Lier, J., Zobel, T., Dahmen, S., Leu, T., Struchtrup, A., Legendre, F., Vesque, C., Schneider-Maunoury, S., Saunier, S., Rüther, U., Gerhardt, C., 2018. Cell type-specific regulation of ciliary transition zone assembly in vertebrates. EMBO J. 37, pii: e97791.

---

## [Author Response]

Essential revisions:1. Data analysis is lacking. A number of figures lack quantitative analysis. Most lack details about the statistical analysis performed.

All information about statistical analysis is now included in figure legends and in the Methods and Materials section.

It is not clear what "#" means in the graphs.

The relevant graphs have been redrawn to make clear the pair-wise comparisons under test

It appears that a simple t-test were used for multiple comparisons without correction.

Statistical tests corrected for multiple comparisons (ANOVA) have been used for Figure 3b and 3d and 6a.

Figure legends need to indicate the n of the experiment and the statistical test used.

We have addressed this point in all figure legends as appropriate

2. Why would increased proteasomal activity (Figure 1C) cause GFP to accumulate in the UbG76V line? It seems that increased proteasomal activity would decrease (or not change) GFP accumulation.

*UbG76V* embryos were treated with a proteasome inhibitor that will prevent degradation of ubiquitinated proteins. Although this procedure was detailed in the Methods and Materials section, we now make this clear in the main text and figure legends, in order to aid interpretation of our observation that loss of Mks1 caused abnormal accumulation of ubiquitinated proteins. Since we also observed a concomitant increase in proteasome activity, we interpret this increase as a response to facilitate the degradation of abnormal amounts of polyubiquitinated proteins. Therefore, when the proteasome is inhibited we observe higher levels of GFP.

3. GST pulldown in Figure 2D is not convincing. The background in the control lane is darker than the experimental lane and could obscure a band of the intensity seen in the GST-UBE2E1 pulldown.

We display a new western blot in Figure 2d from a separate biological replicate of this pulldown

4. Figure 3b: There is little or no difference between the "no colocalization" and "colocalization" images. There may be a quantitative difference in the centrosomal staining seen in the panels in 3a but this would need to be measured.

We have increased the size of magnified insets in Figure 3b and 3d to make the punctate staining at basal bodies more clear. We have attempted to calculate co-localization values between the red (UBE2E1) and green (MKS1) channels (using z-stacks of these channels), but this has been difficult to compare between biological replicates for the two cell-lines because of the variable staining pattern of UBE2E1 in nuclei. We therefore feel that a simple scoring based on presence or absence of UBE2E1 at basal bodies is the most representative of the data.

5. Figure 5c. Western blots need to be added. Why is the level of nuclear Ube2e1 relevant to the work?

We have included the western blots in Figure 5c as requested. We have analysed the overall levels of UBE2E1 localization, including that localized at nuclei, in the “staining intensity” data because it is consistent with the overall increase of protein levels that we observe following western blotting of whole cell extracts. Both data support our contention that MKS1 and UBE2E1 protein levels are co-regulated.

6. The conclusion that increased expression of UBE2E1 decreases MKS1 levels (5d) is not convincing.

In Figure 5d, we have used a clearer western blot from a separate biological replicate of this experiment. (The full blot is included in Suppl. Figure 6). This experiment was conducted three times with consistent results that all demonstrate the co-dependency of protein levels between MKS1 and UBE2E1.

7. The details of what was done in Figure 6b are lacking but a number of points are confusing. Why doesn't the band marked "MKS1cmyc-Ub"show up in the Ub western blot?

The details of methodology for the assay shown in Figure 6b are now included in the Materials and Methods and in the figure legend. In response the second point, the pull-downs bring down all ubiquitinated proteins (the agarose beads are cross-linked to four separate ubiquitin-binding domains) which will be detected by the anti-ubiquitin antibody. Consequently, the specific “MKS1cmyc-Ub” band will be lost in the smear of all ubiquitinated proteins.

Would we expect all ubiquitinated proteins to be detected in the Ub blot? If so, then this blot is not evidence of K63 polyubiquitination of the MKS1.

We agree that all ubiquitinated proteins will be detected in the Ub blot, but this blot is included (in the bottom panel of Figure 6b) as an essential control to show that the K63-specific de-ubiquitinating enzyme (DUB) is active and that the TUBE assay has worked. The top panel of Figure 6b with the anti-cmyc blot shows decreased levels of MKS1 after K63-DUB treatment in comparison to TBST (100 and 149 vs 80 and 121 AUs), suggesting that MKS1-cmyc is at least partially polyubiquitinated with K63 ubiquitin chains.

The differences between the lanes in the c-myc blot are not convincing. I would expected the DUBs to reduce the extent of polyubiquitination, which should have brought the smears of poly-Ub-MKS1cmyc down the gel, but this is not seen.

We now include normalised band intensities for polyUb-cmycMKS1 under the blots in the top panel of Figure 6b, which shows modest yet consistent decreases in polyubiquitination after DUB and K63DUB treatments, for multiple biological replicates of this experiment. However, we agree that de-ubiquitination was not efficient, even following the manufacturer’s protocol, meaning that quantitation between replicates has been problematic. We therefore provide data in Figure 6b as representative of this set of experiments.

8. Figure 6c is problematic. First, both MKS1-GST and Ub-GST are used and a GST western blot is used as evidence of Ub ligation onto MKS1. Ub needs a different tag from its substrate to be convincing.

We agree with this important point, and have gone to considerable efforts to repeat the entire experiment using untagged Ub and His-tagged MKS1 (now displayed in new Figure 6c). This new data shows both mono-ubiquitination of MKS1 and poly-ubiquitination of β-catenin that is dependent on MKS1. (The old data is also shown in Figure 6—figure supplement 1a since it is consistent with these conclusions).

Second, the patterns of the proteins in the blots do not match the guide at the top. Rnf34-His is found in lanes where it was not added.

We apologize for this error in labelling the lanes and have corrected the figure.

UbE2E1-Flag is only seen in the one of the lanes where it is supposed to be present.

We have replaced the input UBE2E1 with a His-tagged protein, and the UBE2E1-FLAG data is included in Figure 6—figure supplement 1a (see above).

This seems to be an unduly complicated figure with UbE2E1-Flag and UbE2E1-His showing different results. There is no explanation of this in the figure legend or text.

We have repeated and simplified the presentation of this entire set of assays (now shown in Figure 6c, with the previous data shown in Figure 6—figure supplement 1a). Figure 6c demonstrates that UBE2E1 undergoes auto-ubiquitination and that ubiquitination is enhanced but not dependent on RNF34. MKS1 inhibits UBE2E1 ubiquitination, suggesting that this could be the basis for the co-dependency of protein levels (data shown in Figure 5c-d). This action is attenuated by addition of β-catenin, but β-catenin by itself does not inhibit UBE2E1 ubiquitination, suggesting that UBE2E1 and MKS1 are co-regulators of β-catenin ubiquitination (also see below, point 9).

9. Figure 6e is too low quality to be interpreted.

This experiment has been repeated and is now shown in Figure 7b. Consistently for all replicates, we observe an increase of polyubiquitinated β-catenin following loss of MKS1 and over-expression of active (wild-type; WT) UBE2E1. This suggests that when MKS1 and active UBE2E1 are present, normal poly-ubiquitination and degradation of β-catenin is taking place. Upon loss of MKS1, this process is disrupted leading to abnormally high levels of highly polyubiquitinated β-catenin. Poly-ubiquitination of β-catenin is much reduced in the presence of inactive (dominant negative; DN) UBE2E1, again suggesting that UBE2E1 and MKS1 are co-regulators of β-catenin ubiquitination.

10. The staining of the neuroepithelium (S2) with the anti-acetylated tubulin antibody shows decreased cytoplasmic staining and no evidence of ciliary label. Better cilia markers, higher magnification and quantification are needed to make this point. This is also a problem in S1D where the MKS mutant cells have strong cytoplasmic label. It is unclear what the arrow is pointing at, but there is no evidence that it is a cilium.

Unfortunately, in our hands, the anti-acetylated tubulin antibody gives variable results on fresh-frozen tissue sections and primary dermal fibroblasts, so we have removed this data in Suppl. Figure 2. We have retained the data showing the increase of activated β-catenin, consistent with data shown in Figure 1a-b.

This is also a problem in S1D where the MKS mutant cells have strong cytoplasmic label.

We agree, but IF staining with this label does not preclude accurate quantitation of cilia incidence and length in these cells and this does seem to be a consistent cellular phenotype in MKS1-mutated cell-lines (see Wheway et al., 2013).

11. It was published that phospho-β-catenin (S33/S37/T41) is degraded by the proteasome at the base of cilia (Corbit et al., 2008; Gerhardt et al., 2015). Szymanska et al., showed that the amount of phospho-β-catenin (S33/S37/T41) is increased by MKS1 deficiency, although proteasome activity is enhanced in MKS1 mutant cells. How can this discrepancy be explained? The authors did not provide a sufficient explanation.

We now make the following potential mechanism clearer in the Discussion: phospho-β-catenin after being phosphorylated needs to be ubiquitinated, probably by UBE2E1 (Chitalia et al., 2008) at the base of the cilium. Our data indicates that MKS1 is necessary to facilitate correct ubiquitination of β-catenin by UBE2E1. Due to the lack of MKS1, UBE2E1 levels increase, which increases polyubiquitination of phosphorylated β-catenin localized at the ciliary base. We observe a concomitant increase in proteasome activity, and suggest that this up-regulation is in response to increased levels of poly-ubiquitinated proteins, including β-catenin, but the levels appear to overload the ability of the proteasome to process them for degradation. As a consequence, there is abnormal accumulation of β-catenin at the ciliary base, where it awaits its degradation, leading to de-regulation of canonical Wnt signalling.

An explanation could be that MKS1 affects Wnt signaling and proteasome activity in two different ways. But this needs to be shown. As they mentioned in the discussion, another transition zone protein, RPGRIP1L, positively regulates proteasome activity (Gerhardt et al., 2015). Previously, it was shown that the loss of RPGRIP1L does not alter the amount of MKS1 at the ciliary transition zone of mouse embryonic fibroblasts (Wiegering et al., 2018) but maybe MKS1 deficiency results in an increased amount of RPGRIP1L at the transition zone of human and mouse fibroblasts leading to the higher proteasome activity the authors observed. Or mutations in MKS1 change the ciliary amount of BBS4 and/or OFD1 which also govern proteasome activity (Liu et al., 2014).

We have addressed this point by using the best available commercial antibody, against RPGRIP1L, and doing a series of western blots following both MKS1 knockdown and in MKS1-mutated fibroblasts (Figure 7—figure supplement 1a-b). RPGRIP1L protein levels were indeed increased in MKS1-mutated fibroblasts compared to normal controls (HDF), but this result was not confirmed by MKS1 knockdown. We therefore conclude that the potential regulatory action of MKS1 on RPGRIP1L warrants further investigation, but is somewhat beyond the current remit and focus of this study.

12. Furthermore, some examinations should be performed in another way to allow a more precise interpretation. For example, the authors quantified phospho-β-catenin (S33/S37/T41) and β-catenin. In MKS1 deficient cells, the amount of both proteins is higher. In my opinion, an important control measurement is missing. The authors should additionally measure the amount of non-phospho-β-catenin (S33/S37/T41).

We have measured levels of non-phospho-β-catenin in both MKS1 knockdown and in MKS1-mutated fibroblasts (Figure 7—figure supplement 1a-b). Β-catenin protein levels were marginally increased following MKS1 knockdown but this result was not confirmed in MKS1-mutated fibroblasts. We conclude that there does not appear to be an overall effect on the levels of non-phospho-β-catenin.

Since Szymanska et al., suggest that MKS1 and UBE2E1 colocalize at the ciliary base, they should also measure the amount of phospho-β-catenin (S33/S37/T41) not only by Western blotting (overall cellular amount) but also at the ciliary base by fluorescence intensity measurements in MKS1 deficient human or mouse fibroblasts.

We present new data in Figure 7—figure supplement 1b to support the data in Figure 7c, showing an increase in phospho-β-catenin at the base cilia but no overall change in non-phospho-β-catenin levels following MKS1 knockdown. We used γ-tubulin staining to define regions-of-interest to ensure that the analyses were limited to the ciliary base; there is no perturbation of γ-tubulin levels following MKS1 knockdown (Figure 7—figure supplement 1b).

This kind of measurement was performed in hTERT-RPE1 cells which were treated with siRNA against MKS1 (Figure 6f) but in these cells the amount of γ-tubulin seems to be increased.

We have confirmed that there is no difference in γ-tubulin levels at the ciliary base following MKS1 knockdown (Figure 7—figure supplement 1a). We have changed the field of view presented in Figure 7c to clarify the observation.

Maybe, the increased amount of phospho-β-catenin (S33/S37/T41) is caused by the higher amount of γ-tubulin. It cannot be excluded that several cilia overlap because this image lacks ciliary axoneme staining.

We agree that this is a possibility, but we have never observed that *MKS1* knock-down or mutant cells have multiple cilia (for example, see Wheway et al., 2013).

In regard to the colocalization of MKS1 and UBE2E1, the authors should perform an analysis such as FRET or in situ PLA which confirms that both proteins localize close enough together to interact at the ciliary base.

We used multiple assays that included Y2H, GST pulldown and IP, and IF microscopy to show that MKS1 and UBE2E1 biochemically interact and co-localise. Importantly, Y2H assays indicate that there is a direct interaction between MKS1 and UBE2E1, and we therefore judge that proximity assays such as FRET and PLA would be merely confirmatory.

13. The proteasome activity assay uses the 20S fluorophore substrate Suc-LLVY-AMC to measure activity. The general view is that the proteasome degrades this substrate independently of ubiquitination (Gan et al., 2019). Despite of this, Szymanska et al., observe an enhanced proteasome activity in cells deficient for UBE2E1, an E2 ubiquitin-conjugation enzyme. This data should be carefully controlled in order to test whether the proteasome activity assay results are reliable. It would be helpful to check whether proteasome activity assays in which other substrates are used that are degraded by the proteasome without ubiquitination provide the same results.

In all relevant assays, we have used proteasome inhibitors to control for its activity. These experiments were conducted in three different cell lines (Figure 1c and d, Figure 4g) always with the control of proteasome inhibitor treatment. In addition, we have used “irrelevant control” cells (mutated for the centrosomal protein ASPM; Figure 1c), which have proteasome activity levels similar to those observed in control cells. We therefore judge that the experiments are well-controlled as presented and that they reflect a reproducible biological effect. Furthermore, in data that we present here, proteasome protein levels appear to be up-regulated, consistent with the increase in activity (see Figure 1c, Figure 7—figure supplement 1b). We therefore do not think that additional proteasome activity assays will provide additional insight above the conventional, accepted assays of activity that we already report.

If this is the case, UBE2E1 may indirectly regulate proteasome activity by taking part in the ubiquitination of a protein that regulates proteasome activity.

We agree that this is a possibility, but another alternative is that loss of UBE2E1 causes dysregulation of another E2 or E2s that cause aberrant increase of polyubiquitinated proteins and hence increase in proteasome activity in degradation of polyubiquitinated proteins. Discriminating between these alternatives is, we feel, beyond the focus of the manuscript.

14. In human there are approximately 40 E2 enzymes, therefore each E2 is likely to act with multiple E3s (there are over a 1000) with E3s likely to be the main driver of target specificity. Authors have not controlled their experiments, or at least have not showed it in the manuscript, with any other E2s. Consequently, there is a real risk that the findings simply confirm that cilia and MKS1 are regulated by UPS rather than that UBE2E1 is a specific regulator in this system.

The specific localization and interactions of UBE2E1, and reciprocal co-regulation of MKS1 and UBE2E1, argue that UBE2E1 is a specific co-regulator. In particular, the dominant negative (DN) mutant C131S of UBE2E1 has specific and significant effects on ciliogenesis. However, we agree that *degradation* of MKS1 (rather than co-regulation) is likely to be mediated by another E2 (for example, Ube2l3, Ube2k, Ube2n or Ube2m; see Figure 4e-f source data 1) because polyubiquitination of MKS1 is inhibited, rather than enhanced, by UBE2E1 (see Figure 6a).

15. With regard to immunoprecipitations and ubiquitination assays. Firstly inputs (for each analyzed protein in each sample) and/or loading controls are not always shown (e.g. Figure 2) which makes interpretation of results difficult.

Input material (whole cell extracts; WCE) and loading controls are now shown for the TUBE assays (Figure 6b and 7b). Input material is also shown for the pulldowns in Figure 2d-f (also see below, point 18).

16. Details are lacking about the conditions of the IP experiments used to detect ubiquitinated proteins (6b,d,e). If these were performed under native conditions, the ubiquitin being detected by western blot could be on any protein that complexes with the target of the precipitation. E2 ligases may have a large number of interactors.

We apologize for this oversight and now provide further details of the methodology used in our IP experiments. We now make clear that we used denaturing conditions for the western blots. We note that TUBE assays (Figure 6b and 7b) pull-down all poly-ubiquitinated proteins and the western blot is then performed to visualize the protein of interest.

17. As far as the UBE2E1 knockdown model. In the MS analysis in Figure 3, I am quite surprised that there is still such a strong recovery of UBE2E1 peptides from UBE2E1 silenced line. Is this not an impediment to this model? Does this not suggest that this cell line still has sufficient, functioning UBE2E1? Also again, it is hard to estimate if the MKS1 specific observations would not be replicated by silencing other E2s.

We think that the reviewers may have misinterpreted what is shown in Figure 4e: the peptide count ratios are for the various different protein species immunoprecipitated by anti-MKS1 from the shScrambled control vs the sh*Ube2e1* cell-lines (with the raw data tabulated in Figure 4e-f source data 1, columns Q-S). We do not observe recovery of Ube2e1 peptides under these conditions, but there are moderate levels of other E2s (Ube2l3, Ube2k, Ube2n and Ube2m). Successful Ube2e1 knock-down in the sh*Ube2e1* cell-line is demonstrated in Figure 4c.

18. Lack of inputs in Figure 6a,b,d and e make interpretation of the critical claims from this manuscript difficult. If there is an effect of UBE2E1 on MKS1 total protein levels inputs of UBE2E1, MKS1 and loading controls for all samples in 6a 6b and 6d should clarify this.

Input material is now shown for Figures 2d-f, 6b-c and 7a-b (also see point 15). Input recombinant proteins are also shown individually for the in vitro ubiquitination assays (Figure 6c, Figure 6—figure supplement 1a).

Currently, the only demonstration of the claimed protein increase is in Figure 5d and it is not repeated 3 times to allow for statistical analysis. Furthermore, it looks very marginal (numbers are not greatly affected either) 113-107-96…

In Figure 5d, we have used a clearer western blot from a separate biological replicate of this experiment (see also response to point 6).

..with appreciation that (WB is not really a quantitative technique) and unlikely to be significant.

As mentioned above (point 6), this experiment was conducted three times with consistent results that all demonstrate the co-dependency of protein levels between MKS1 and UBE2E1. We have not provided a statistical analysis of the replicates for this assay exactly for the reasons stated by the reviewer for this assay (the ECL method for western blot visualization is difficult to optimize for semi-quantitative purposes).

It is possible that even if direct, changes caused by UBE2E1 to MKS1and vice versa are completely irrespective of protein levels or stability and affect localization and function through chains other than K48.

The TUBE data shown in Figure 6b suggests that MKS1 is modified by K63 ubiquitination. We have also attempted to confirm this directly by LC-MS/MS analysis of peptide species, but unfortunately, in our hands, the B9 domain appears to be particularly resistant to standard trypsin digestion and even the trypsin/Lys-C protocol for tightly folded proteins does not provide coverage of the predicted modified lysine groups in MKS1.

19. Authors should clarify statements regarding degradation and UBE2E1 dual function, as alternative explanations have not been ruled out. UBE2E1 action on MKS1 could be indirect.

The interpretation of our data is that the direct interaction between MKS1 and UBE2E1 mediates K63 mono/bi-ubiquitination of MKS1, modulating MKS1 function, ciliogenesis and other known downstream functions such as β-catenin polyubiquitination. However, we assume that another E2 or E2s regulate degradation/polyubiquitination of MKS1.

The increase in b-catenin levels in Figure 6d is very hard to interpret without loading controls.

To aid interpretation of our data, we have provided loading controls and a new TUBE assay has been performed (Figure 7b).

20. There are multiple alternative explanations for the observations presented in this manuscript, which renders figure 7 a bit uncertain. For example, UBE2E1 could be involved in ubiquitination of an MKS1 targeting E3 or DUB and that could be the reason for the observed changes. It would be more forthcoming to highlight this in the manuscript.

In response to point 19 above, we now provide alternative interpretations of our data in the Discussion, but the most parsimonious explanation of our findings are summarized in Figure 7d. in the main text we now state: “…alternative interpretations of our data are that UBE2E1 could occupy an E2 binding position on MKS1 preventing MKS1 degradation, or that MKS1 binds to UBE2E1 to facilitate UBE2E1 ubiquitination and degradation”